# Enhanced BRAF engagement by NRAS mutants capable of promoting melanoma initiation

Brandon M. Murphy[1], Elizabeth M. Terrell[2], Venkat R. Chirasani[3], Tirzah J. Weiss[1], Rachel E. Lew[1,4], Andrea M. Holderbaum [1,4], Aastha Dhakal[4], Valentina Posada[4], Marie Fort[4], Michael S. Bodnar[1,4], Leiah M. Carey[3], Min Chen[1,5], Craig J. Burd[4], Vincenzo Coppola [1,5], Deborah K. Morrison [2], Sharon L. Campbell [3] & Christin E. Burd [1,4✉]

A distinct profile of NRAS mutants is observed in each tumor type. It is unclear whether these profiles are determined by mutagenic events or functional differences between NRAS oncoproteins. Here, we establish functional hallmarks of NRAS mutants enriched in human melanoma. We generate eight conditional, knock-in mouse models and show that rare melanoma mutants (NRAS G12D, G13D, G13R, Q61H, and Q61P) are poor drivers of spontaneous melanoma formation, whereas common melanoma mutants (NRAS Q61R, Q61K, or Q61L) induce rapid tumor onset with high penetrance. Molecular dynamics simulations, combined with cell-based protein–protein interaction studies, reveal that melanomagenic NRAS mutants form intramolecular contacts that enhance BRAF binding affinity, BRAF-CRAF heterodimer formation, and MAPK > ERK signaling. Along with the allelic series of conditional mouse models we describe, these results establish a mechanistic basis for the enrichment of specific NRAS mutants in human melanoma.

[1] Department of Cancer Biology and Genetics, The Ohio State University, Columbus, OH 43210, USA. [2] Laboratory of Cell and Developmental Signaling, National Cancer Institute-Frederick, Frederick, MD 21702, USA. [3] Department of Biochemistry & Biophysics and Lineberger Comprehensive Cancer Center, University of North Carolina at Chapel Hill, Chapel Hill, NC 27599, USA. [4] Department of Molecular Genetics, The Ohio State University, Columbus, OH 43210, USA. [5] Genetically Engineered Mouse Modeling Core, The Ohio State University, Columbus, OH 43210, USA. ✉email: burd.25@osu.edu

It is unclear why the profile of oncogenic *RAS* mutations differs between tumor types. It was once thought that differences in tumor etiology determined the preferred location (codon 12, 13, or 61) and amino acid identity of oncogenic mutations in *RAS*. However, apart from $KRAS^{12C}$ mutations which are linked to cigarette carcinogens in lung cancer[1], tumor type-specific mutational processes do not explain the enrichment of specific *RAS* mutations in many cancers. This trend is particularly evident in melanoma where the most common *NRAS* mutations (Q61R and Q61K) are not caused by direct damage from ultraviolet (UVB) light[2]. These observations suggest that each RAS mutant may fulfill different requirements for tumor initiation.

Emerging evidence shows that RAS mutants have distinct biochemical and tumorigenic properties. While all oncogenic RAS mutants are constitutively active, differential positioning of the switch I and II domains leads to variations in GTP binding and hydrolysis[3]. These structural differences can also influence effector interactions as evidenced by the positioning of switch II in $KRAS^{12R}$, which prevents PI3Kα binding and the subsequent induction of micropinocytosis[4,5]. Such mechanistic differences may also explain the tissue-specific potential of RAS mutants to initiate tumorigenesis in genetically engineered mouse models (GEMMs). For example, we have shown that endogenous levels of $NRAS^{61R}$ or $NRAS^{12D}$ exhibit distinct tumorigenic potential in GEMMs of melanoma and leukemia[6]. Finally, mutation-specific functions of oncogenic RAS may influence patient outcomes as the efficacy of targeted therapies in colorectal and non-small cell lung cancer is dependent upon the underlying KRAS mutant[7–9]. Therefore, understanding functional differences that drive the selection of specific RAS mutants in each cancer type may identify pharmacologically tractable targets required for tumor initiation.

Technical challenges have made it hard to identify differences between *RAS* alleles that drive tumorigenesis. For example, exogenous gene expression is a commonly used tool, yet *RAS* gene dosage has been shown to affect signaling[10], localization[11] and in vivo functionality[12,13]. The biological consequences of mutant RAS expression also differ based on the isoform (H-, K- or N-RAS) and cell-type examined[6,14–17]. Therefore, it is essential to assess the differences between endogenous RAS mutants under physiologically relevant conditions.

Here, we report the development of eight NRAS-mutant mouse alleles, each of which enables the conditional expression of a distinct NRAS mutant from the endogenous gene locus. Crossing these alleles to a melanocyte-specific Cre, we find that the melanomagenic potential of NRAS mutants parallels their frequency in human melanoma. We link the melanomagenic potential of NRAS mutants to enhanced BRAF binding, dimerization, and MAPK > ERK signaling.

## Results

### The tumorigenic potential of NRAS mutants parallels allelic frequency in human melanoma.

We used CRISPR-Cas9 to zygotically modify the *Nras* mutation in Tyr::CreER$^{T2}$; *LSL-Nras$^{61R/R}$* ($TN^{61R/R}$) mice (Supplementary Figs. 1a, b, 2a; refs. [6,18]). This process yielded eight mouse models in which induction of Cre recombinase triggers the melanocyte-specific expression of a modified *Nras* gene from the endogenous locus: $TN^{61K/K}$, $TN^{61L/L}$, $TN^{61H/H}$, $TN^{61P/P}$, $TN^{61Q/Q}$, $TN^{12D/D}$, $TN^{13D/D}$, and $TN^{13R/R}$. Each *LSL-Nras* allele was sequenced and functionally validated in mouse embryonic fibroblasts (MEFs) (Supplementary Figs. 1c–e, 2b–d). Founder animals were backcrossed two generations to $TN^{61R/R}$ mice to limit any off-target effects of CRISPR-Cas9.

We used this suite of *TN* mice to determine if *NRAS* oncogenes common to human melanoma (Fig. 1a) drive melanocyte transformation better than those present in other tumor types. Experimental $TN^{61X/X}$ cohorts were generated by intercrossing Tyr::CreER$^{T2}$ transgenic mice carrying one *LSL-Nras$^{61R}$* and one *LSL-Nras$^{61X}$* allele, where X = K, L, H, P, or Q (Supplementary Fig. 1f). The resulting offspring were topically treated with 4-hydroxytamoxifen (4-OHT) on postnatal days 1 and 2 to drive CreER$^{T2}$-mediated excision of the *LSL* transcriptional stop sequence and initiate expression of each *Nras* variant (Supplementary Fig. 1g). The mice were then subjected to a single, 4.5 kJ/m$^2$ dose of ultraviolet B (UVB) irradiation on postnatal day 3 to mimic the role of sunlight in melanoma formation (Supplementary Fig. 1g; ref. [18]).

Spontaneous melanomas formed more rapidly and frequently in $TN^{61R/R}$ and $TN^{61K/K}$ mice than in $TN^{61L/L}$ or $TN^{61H/H}$ animals, and no tumors were detected in the $TN^{61P/P}$ and $TN^{61Q/Q}$ models (Fig. 1b, c; Supplementary Table 1a). These differences were not due to litter-specific effects as the onset, burden, and growth rates of $TN^{61R/R}$ tumors did not differ between experimental cohorts of male and female mice (Supplementary Fig. 1h–l; Supplementary Tables 1b, c). Melanoma growth rates, measured with digital calipers, were similar regardless of genotype (Fig. 1d; Supplementary Table 1a), leading to overall survival rates which paralleled the tumor onset for each $TN^{61X/X}$ model (Supplementary Fig. 1m). Immunohistochemistry (IHC) staining of tumor sections with a Ki67 antibody showed that proliferation rates were slightly higher in $TN^{61R/R}$, $TN^{61K/K}$ and $TN^{61L/L}$ than in $TN^{61H/H}$ melanomas (Supplementary Fig. 1n; Supplementary Table 1d). IHC staining for CD45$^+$ cells or cleaved Caspase indicated no difference in immune infiltration or apoptosis among tumors of different NRAS genotypes (Supplementary Fig. 1o, p; Supplementary Table 1d). UVB light cooperated equally with each NRAS mutant to enhance tumor onset and burden, revealing that differences in the melanoma-driving capabilities of each variant are independent of UVB carcinogenesis (Supplementary Fig. 3a–d; Supplementary Table 1e).

Our results in the $TN^{61X/X}$ models and the rarity of codon 12/13 mutants in human melanoma suggested that $TN^{12D/D}$, $TN^{13D/D}$, and $TN^{13R/R}$ mice would not develop tumors. To test this hypothesis, we generated experimental colonies by breeding mice homozygous for each codon 12 or 13 allele in our series. $TN^{12D/D}$ and $TN^{13D/D}$ mice did not succumb to melanoma after 60 weeks of observation (Supplementary Fig. 2e, f). By contrast, $TN^{13R/R}$ mice did form melanomas, albeit with lower efficiency than the weakest melanoma-forming codon 61 model, $TN^{61H/H}$. These data, summarized in Supplementary Table 1f, establish differences in the ability of oncogenic NRAS mutants to initiate melanoma formation and provide a plausible explanation for the prevalence of $NRAS^{61R}$ and $NRAS^{61K}$ mutations in human melanoma.

### NRAS proteins with compromised GTPase activity facilitate $NRAS^{61R}$-dependent melanomagenesis.

In RAS-driven malignancies, the complementary wild-type allele is thought to suppress tumorigenesis driven by the mutationally-active oncoprotein[19–22]. However, the function of wild-type RAS may be mutation-specific as the presence of wild-type KRAS results in the selection of $KRAS^{61R}$ over $KRAS^{61L}$ in urethane-induced, murine lung tumors[23]. In melanoma, the effect of wild-type NRAS on the tumorigenic potential of mutant NRAS is unclear. Further studies examining the interaction between alleles of differing oncogenic potential could shed light on the functional interplay between RAS molecules. To explore the interaction between *NRAS* alleles in melanocytes, we compared the tumor

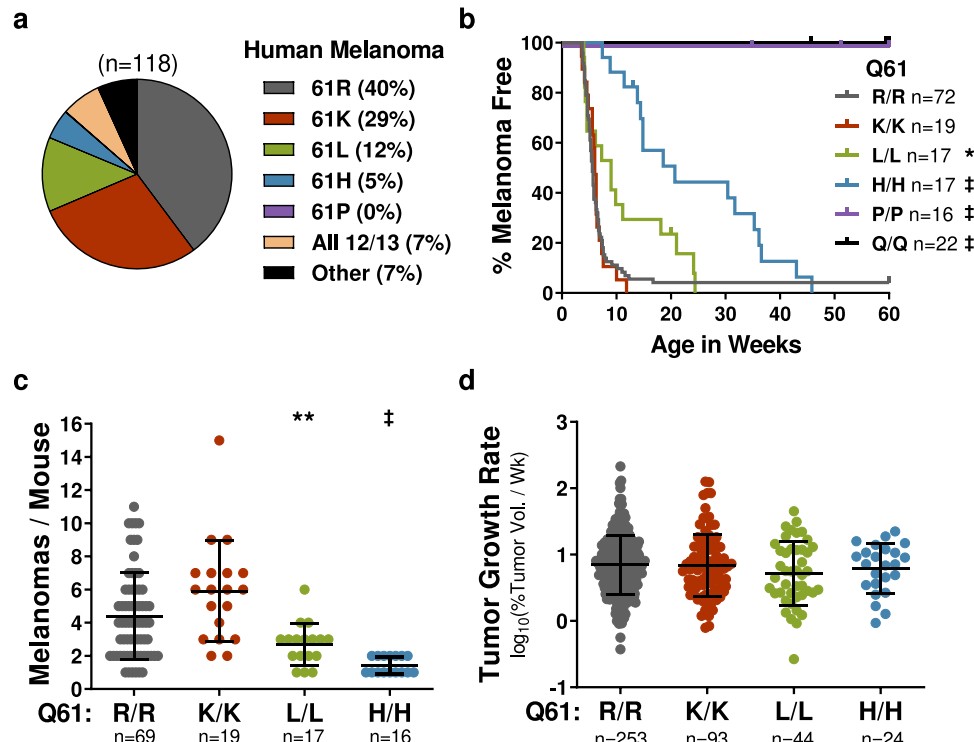

**Fig. 1 Frequency of NRAS mutants in human melanoma parallels tumorigenic potential in mice. a** Frequency of *NRAS* mutations in the TCGA PanCancer Atlas dataset for human cutaneous melanoma. Melanoma-free survival (**b**), total tumor burden (**c**), and tumor growth rates (**d**) for mice expressing the indicated melanocyte-specific NRAS mutants. Tumor burden and growth rate data are presented as mean values $+/-$ SD. The following number biologically independent animals were evaluated per genotype (61R = 72, 61K = 19, 61L = 17, 61H = 17, 61P = 16, 61Q = 22). Log-rank (Mantel–Cox) (**b**) or ANOVA (**c**, **d**) with a Tukey's multiple comparisons test was used to compare measurements between each genotype. $TN^{X/X}$ samples statistically different from $TN^{61R/R}$ are indicated in the figure. Adjusted *p*-values for all comparisons can be found in Supplementary Table 1a. * $p < 0.05$, ** $p < 0.01$, ‡ $p < 0.0001$. Source data are provided as a Source Data file.

onset, burden, and overall survival of homozygous ($TN^{61R/R}$, $TN^{61X/X}$) and heterozygous ($TN^{61X/R}$) mice from each of our experimental cohorts. The melanoma phenotypes of heterozygous mice were generally intermediate to those observed in homozygous $TN^{61R/R}$ and $TN^{61X/X}$ animals from the same cohort (Fig. 2a–d; Supplementary Table 1e). Even though loss-of-heterozygosity is uncommon in NRAS-driven human malignancies (Supplementary Fig. 4a–c), NRAS$^{61R}$ was unable to drive melanoma formation in the presence of wild-type NRAS$^{61Q}$ (Fig. 2e). These data reveal that the additive effect of *Nras* alleles in melanoma is dependent upon the loss of intrinsic GTPase activity.

**Melanomagenic NRAS mutants drive transcriptional profiles associated with proliferation.** We performed RNA sequencing on MEFs derived from our melanomagenic (61R/R and 61H/H) and non-melanomagenic (61P/P and 61Q/Q) *TN* models to identify transcriptional profiles downstream of each NRAS mutant. Transcriptomes elicited by the melanomagenic NRAS$^{61R}$ and NRAS$^{61H}$ mutants clustered separately from those elicited by the non-melanomagenic NRAS$^{61P}$ and NRAS$^{61Q}$ mutants in principal component analysis (Supplementary Fig. 5a). To identify the major determinants of these clusters, we first compared the transcriptomes of $TN^{61R/R}$, $TN^{61H/H}$, or $TN^{61P/P}$ MEFs to wild-type, $TN^{61Q/Q}$ MEFs (Supplementary Fig. 5b–e; Supplementary Data 1a–c). As expected, transcripts associated with E2F and MYC were enriched in MEFs expressing mutant NRAS (Supplementary Fig. 5f–h); however, this enrichment was most pronounced in $TN^{61R/R}$ MEFs. Together, these data suggest a

potential link between melanomagenic *NRAS* alleles and enhanced proliferative signaling.

We next sought to identify mutant-specific transcriptional programs by comparing the transcriptomes of MEFs expressing different NRAS mutants. Only 23 transcripts differed between MEFs expressing either melanomagenic NRAS mutant (61R/R and 61H/H; ≥1.5-fold, *p*-adj < 0.05) (Supplementary Data 2a). However, at least 922 transcripts differed between MEFs expressing melanomagenic and non-melanomagenic NRAS mutants (Supplementary Data 2b, c). Gene ontology (GO) analysis identified gene sets associated with GTPase activation and guanyl nucleotide binding as top biological processes enriched in $TN^{61R/R}$ and $TN^{61H/H}$, over $TN^{61P/P}$ MEFs (Fig. 3a). Gene set enrichment analysis (GSEA) further revealed that transcripts enriched by melanomagenic NRAS mutants were those associated with MYC and KRAS signaling, including feedback inhibitors of the MAPK pathway (e.g., *DUSP6*, *SPRY2*) (Fig. 3b, c). Of relevance, heightened MAPK > ERK signaling drives the expression of these feedback inhibitors[24] and *DUSP6* and *SPRY2* levels are elevated in human skin cancers (Supplementary Fig. 6a). Subsequent qRT-PCR experiments confirmed that these transcripts were also elevated in immortalized $TN^{61R/R}$ and $TN^{61H/H}$ melanocytes, suggesting that mutant-specific transcriptional profiles are conserved between cell types (Supplementary Fig. 6b). Despite upregulation of these inhibitors, proliferation, as measured by EdU incorporation, was higher in MEFs and cutaneous melanocytes expressing melanomagenic NRAS mutants than those expressing non-melanomagenic NRAS mutants (Fig. 3d–f; Supplementary Fig. 7; Supplementary Table 2).

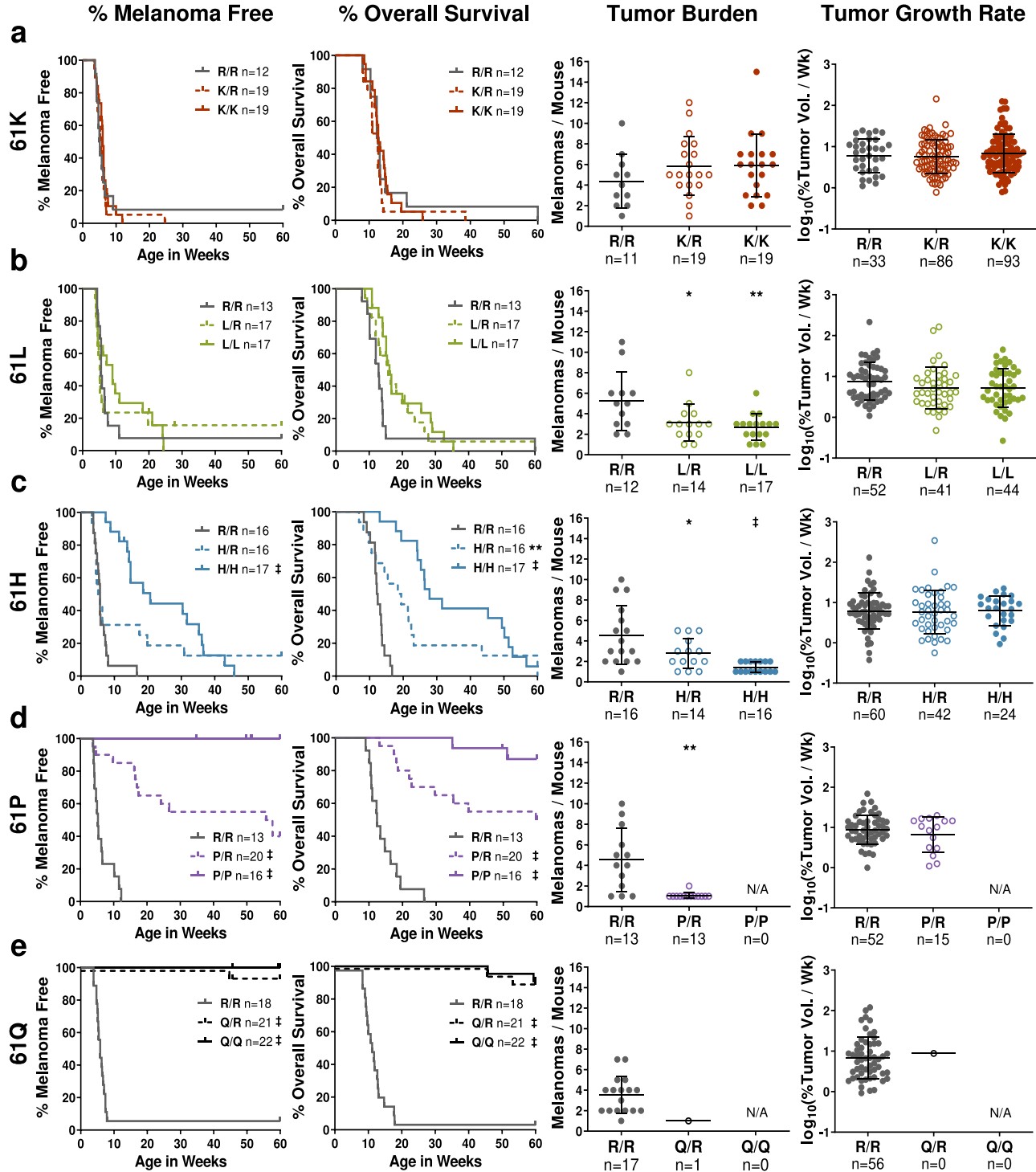

**Fig. 2 Combining codon 61 mutants defective in GTPase activity results in an intermediate melanoma phenotype.** Melanoma-free survival, overall survival, tumor burden, and tumor growth rates for the following treatment cohorts: **a** $TN^{61K/R}$, **b** $TN^{61L/R}$, **c** $TN^{61H/R}$, **d** $TN^{61P/R}$, and **e** $TN^{61Q/R}$. Tumor burden and growth rate dot plots are presented as mean values +/− SD. The following number biologically independent animals were evaluated per genotype (61K cohort: R/R = 12, K/R = 19, K/K = 19; 61L cohort: R/R = 13, L/R = 17, L/L = 17; 61H cohort: R/R = 16, H/R = 16, H/H = 17; 61P cohort: R/R = 13, P/R = 20, P/P = 16; 61Q cohort: R/R = 18, Q/R = 21, Q/Q = 22). In **a**–**e**, the phenotype of $TN^{61R/R}$ mice was compared to $TN^{61X/X}$ and $TN^{61X/R}$ animals. Log-rank (Mantel–Cox) tests were used to compare survival. One-way ANOVA with a Dunnet T3 multiple comparisons test was used to compare tumor burden and growth between each genotype and $TN^{61R/R}$ for that cohort. Adjusted $p$-values for all comparisons can be found in Supplementary Table 1f. * $p < 0.05$, ** $p < 0.01$, ‡ $p < 0.0001$. Source data are provided as a Source Data file.

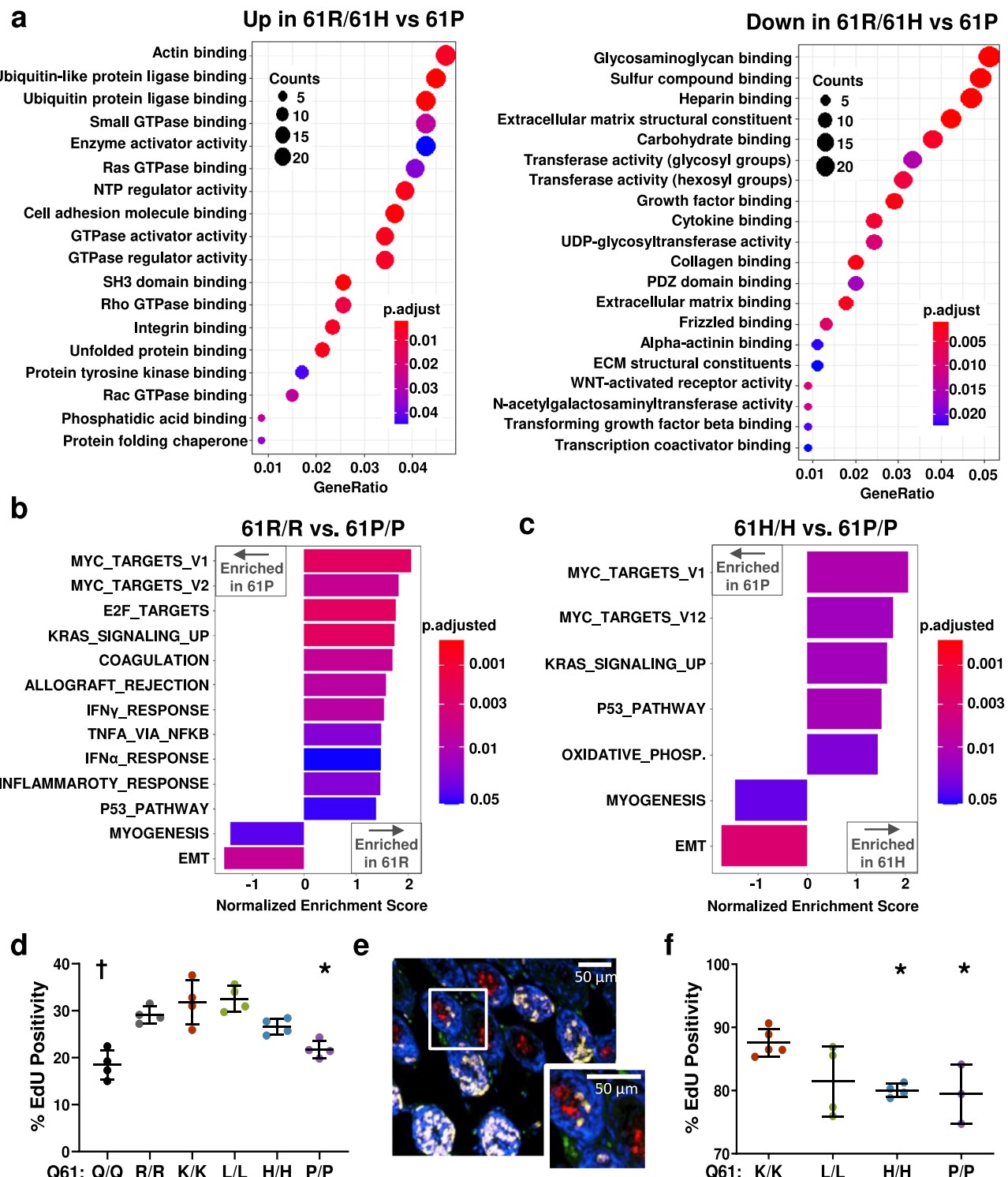

**Fig. 3 Differential regulation of the RAS-Myc axis by melanomagenic and non-melanomagenic NRAS mutants. a** Dot plots representing the molecular functions subset of Gene Ontology (GO) analysis of genes upregulated (left) or downregulated (right) in $TN^{61R/R}$ and $TN^{61H/H}$ MEFs compared to $TN^{61P/P}$ MEFs. Three biological replicates per genotype were used for analyzed. Bar plot showing the differential enrichment of Hallmark gene sets ($p$-adjusted < 0.05) in MEFs expressing NRAS$^{61R/R}$ versus NRAS$^{61P/P}$ (**b**) or NRAS$^{61H/H}$ versus NRAS$^{61P/P}$ (**c**). **d** Dot plot of flow cytometric analysis of EdU labeling in NRAS-mutant MEFs. $n = 4$ biologically independent MEF lines per genotype were examined over 4 independent experiments. **e** Representative image of EdU (proliferation, green) and gp100 (melanocyte, red) co-staining in skin harvested from a ten-day old mouse. $n = 4$ biologically independent mice were examined per genotype. **f** Dot plot of percent EdU positivity in melanocytes from 10-day old $TN^{61X/X}$ mouse skin. The following number biologically independent animals were evaluated per genotype (K/K = 5, L/L = 4, H/H = 4, P/P = 3). Dot plot data are presented as mean values +/− SD where each dot represents one biological replicate. One-way ANOVA with a Tukey's post-test was used to compare data between each genotype. NRAS mutant samples statistically different from NRAS$^{61R/R}$ samples are indicated in the figure. Adjusted $p$-values for all comparisons can be found in Supplementary Table 2. * $p < 0.05$, † $p < 0.001$. Source data are provided as a Source Data file.

To determine if the transcriptional effects we observed would persist following tumor onset, we performed RNA-sequencing on spontaneous melanomas isolated from our *TN* GEMMs. Fewer than 35 genes were differentially expressed between tumors carrying NRAS mutants considered to be strong melanoma drivers (Supplementary Fig. 8a, b; Supplementary Data 3a, b). Conversely, 761 genes were differentially expressed between tumors expressing a strong (61R) and a weak (61H) driver of melanoma initiation (Supplementary Fig. 8c; Supplementary Data 3c). When compared to *TN$^{61H/H}$* melanomas, *TN$^{61R/R}$* melanomas were enriched for transcripts associated with immune pathway regulation (Supplementary Fig. 8d). However, a decrease in immune infiltration was not consistent among end-stage *TN$^{61H/H}$* melanomas, when surveyed by IHC for CD45$^+$ (Supplementary Fig. 1o). Furthermore, GO analysis paralleled in vitro MEF data identifying processes associated with guanyl nucleotide binding as top biological hits enriched in *TN$^{61R/R}$* tumors over *TN$^{61H/H}$* tumors (Supplementary Fig. 8e). These data suggest that higher RAS activity and proliferative signaling are common functions of melanomagenic NRAS mutants maintained throughout tumorigenesis.

**Melanomagenic NRAS mutants promote MAPK > ERK signaling**. To test the idea that MAPK signaling is elevated in the presence of melanomagenic NRAS mutants, we analyzed ERK and AKT activation in MEFs and immortalized melanocytes from our *TN$^{61X/X}$* models. We induced NRAS expression in each cell type using adenoviral Cre, allowed the cells to recover from infection, and then placed the cells in serum-free media for 4 h prior to protein isolation. Phospho-ERK levels paralleled the melanomagenic potential of NRAS mutants in both MEFs and melanocytes (Fig. 4a; Supplementary Figs. 9a, b, 10a; Supplementary Table 3a, b). However, activation of the PI3K/AKT signaling pathway did not parallel melanomagenicity in either MEFs or melanocytes from our *TN* models (Fig. 4a; Supplementary Figs. 9a, c, 10a). These differences in NRAS signaling appeared to persist throughout tumorigenesis as an analogous pattern of mutant-specific MAPK > ERK, but not PI3K/AKT, signaling was observed in melanomas from our *TN* models (Fig. 4b and Supplementary Table 3a). Together, these results link the melanomagenic potential of NRAS mutants to enhanced MAPK > ERK signaling.

**Melanomagenic NRAS mutants promote RAF dimerization**. Mutationally-active RAS proteins stimulate signaling through the RAF > MEK1/2 > ERK1/2 pathway using both direct and indirect mechanisms. Mutant RAS can indirectly activate MAPK through the allosteric regulation of SOS1, which in turn promotes GTP loading on wild-type RAS isoforms[25]. To determine if melanomagenic NRAS mutants promote higher levels of MAPK signaling via this indirect mechanism, we used lentiviral shRNAs to knockdown *Sos1* or *Hras* and *Kras* in *TN$^{61R/R}$*, *TN$^{61P/P}$* and *TN$^{61Q/Q}$* MEFs. Knockdown of *Nras* served as a positive control and reduced MAPK pathway activation in MEFs expressing NRAS$^{61R/R}$ (Fig. 5a). However, knockdown of *Sos1* or *Hras* and *Kras* had no effect on MAPK activation regardless of the *Nras* allele present, ruling out the possibility that melanomagenic NRAS mutants drive heightened MAPK signaling through the indirect activation of wild-type RAS (Fig. 5a; Supplementary Table 4a).

RAS isoforms and KRAS mutants have distinct affinities for each RAF homolog in exogenous expression systems[26]. Thus, we postulated that melanomagenic NRAS mutants might activate RAF better than non-melanomagenic mutants in our endogenous expression system. Knockdown of *Braf* or *Craf* using lentiviral

shRNA partially reduced MAPK activation in *TN$^{61R/R}$*, *TN$^{61P/P}$*, and *TN$^{61Q/Q}$* MEFs (Fig. 5b; Supplementary Table 4b). *Araf* knockdown, by contrast, enhanced ERK activation in *TN$^{61R/R}$* MEFs (Fig. 5b). To confirm these results, we developed an adenoviral NanoBiT system to measure RAF homo- and heterodimerization in live cells (Fig. 6a). We induced NRAS expression in MEFs and primary melanocytes from each *TN* model and then infected the cells with adenovirus encoding BRAF-LgBiT and BRAF-SmBiT, BRAF-LgBiT and CRAF-SmBiT, CRAF-LgBiT and CRAF-SmBiT, ARAF-LgBiT and BRAF-SmBiT, ARAF-LgBiT and ARAF-SmBiT, or ARAF-LgBiT and CRAF-SmBiT. Elevated BRAF-BRAF and BRAF-CRAF dimers were consistently observed in MEFs and primary melanocytes expressing NRAS mutants with strong melanoma-driving potential (Figs. 6b–g; Supplementary Figs. 10b–d, 11a–f; Supplementary Table 4c–e). These results show that the ability of NRAS mutants to drive melanoma in vivo parallels the induction of BRAF dimers in vitro.

**Melanomagenic NRAS mutants bind BRAF with greater affinity**. We hypothesized that melanomagenic NRAS mutants adopt structural conformations that promote BRAF binding and dimerization. To test this hypothesis, we performed molecular dynamics (MD) simulations to predict the most common conformers of NRAS Q61-R, -K, -H, -L, and -P. As protein conformational sampling using traditional MD simulations is limited by high energy barriers during structural transitions, we employed Replica-exchange molecular dynamics (REMD[27],) simulations to enhance conformational sampling. More intramolecular contacts with the mutant amino acid side chain were observed in prominent conformers of the melanomagenic NRAS mutants than in the non-melanomagenic NRAS mutants (Fig. 7a; Supplementary Fig. 12a–e). These intramolecular interactions were predicted to alter the conformation and dynamic properties of the switch I and II regions. Because the conformation of switch I and II influences RAS effector binding[28], we performed HEX docking simulations to test how well each NRAS mutant bound to full-length BRAF (Fig. 7b). The most frequently sampled conformers of NRAS$^{61R}$ and NRAS$^{61K}$ bound BRAF with the highest affinity, followed by the third most common mutant in human melanoma, NRAS$^{61L}$ (Fig. 7b). These findings suggested that melanomagenic codon 61 substitutions may stabilize NRAS conformations with increased BRAF binding affinity.

To test whether melanomagenic NRAS mutants have enhanced BRAF affinity in vivo, we performed cell-based bioluminescence resonance energy transfer (BRET) assays. In these systems, the close proximity of BRAF molecules fused to the donor, Rluc8, and NRAS molecules fused to the acceptor, Venus, leads to BRET (i.e., fluorescence at 528 nm). By holding the amount of transfected energy donor (BRAF) constant and increasing the amount of acceptor (NRAS), the relative binding affinity (BRET$_{50}$) of each NRAS-BRAF pair can be determined. Strong initiators of melanoma, like NRAS$^{61R}$ and NRAS$^{61K}$, showed higher BRAF affinity (lower BRET$_{50}$) than weaker and non-melanogenic alleles like NRAS$^{61H}$, NRAS$^{61P}$, and NRAS$^{12D}$ (Fig. 7c). Consistent with our NanoBit data, CRAF affinity did not differ between melanomagenic and non-melanomagenic NRAS mutants (Fig. 7d). Together, these findings put forth a model in which melanomagenic NRAS substitutions stabilize protein conformations with high BRAF affinity, leading to increased RAF dimerization, MAPK > ERK signaling, and melanocyte transformation (Fig. 8).

## Discussion
Here we establish that functional differences underlie the enrichment of specific NRAS mutants in human melanoma.

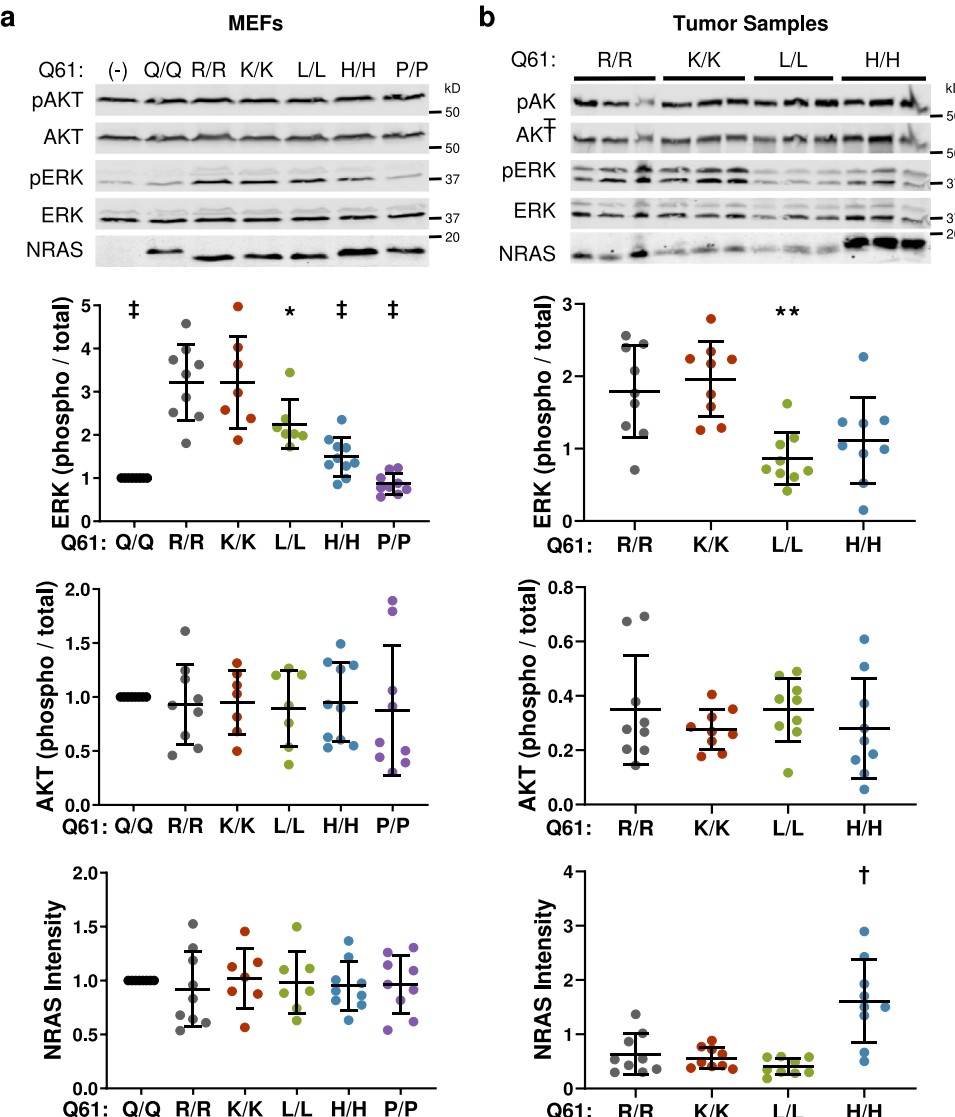

**Fig. 4 MAPK pathway activation parallels the tumorigenic potential of oncogenic NRAS mutant.** Immunoblot of protein lysates isolated from MEFs (**a**) or murine melanomas (**b**) expressing the indicated NRAS mutants. Dot plots showing the quantification of ERK activation, AKT activation, or NRAS expression. Dot plot data are presented as mean values +/− SD where each dot represents one biological replicate. For **a** following number biologically independent replicates per genotype were examined over nine independent experiments (Q/Q = 9, R/R = 9, K/K = 7, L/L = 7, H/H = 9, P/P = 9). For **b** nine biologically independent replicates were assessed per genotype. One-way ANOVA with a Tukey's post-test was used to compare data between each genotype. NRAS mutant samples statistically different from NRAS[61R/R] samples are indicated in the figure. Adjusted $p$-values for all comparisons can be found in Supplementary Table 3a. * $p < 0.05$, ** $p < 0.01$, † $p < 0.001$. Source data are provided as a Source Data file.

Previous publications highlight differences in the tumorigenic potential of *RAS* codon 12 and 61 mutations in pancreatic cancer, lung cancer, leukemia, and melanoma[6,17,29,30]. However, these results might be predicted because codon 61 mutants have a more profound effect on RAS intrinsic GTPase activity[31,32]. What remained unclear is why certain codon 61 mutants would be more prevalent than others in melanoma. We explored this question in a suite of eight GEMMs and discovered a direct correlation between the frequency of a particular NRAS mutant in human melanoma and its melanomagenic potential in mice (Fig. 1b, c). Thus, functional differences among the NRAS oncoproteins, rather than preferential UV carcinogenesis, determines which NRAS mutants occur in human melanoma. This discovery opens the door for therapeutic and preventative strategies targeting functions exclusive to melanomagenic NRAS mutants.

Our analysis of heterozygous *TN* mice revealed an interesting, additive effect of mutant, but not wild-type, NRAS on melanomagenesis. Prior studies show that a single, wild-type *RAS* allele can limit the tumorigenic potential of RAS mutants of the same isoform[33]. This observation is supported by data from several human tumor types in which loss or downregulation of the cognate wild-type allele is frequent[19–21]. In line with these findings, our results reveal that NRAS[61R] cannot initiate melanoma formation in the presence of a wild-type allele (Fig. 2e). However, NRAS[61R] retains the ability to initiate melanoma when expressed in combination with a non-melanomagenic, GTPase defective NRAS[61P] allele (Fig. 2d). *NRAS* loss of heterozygosity (LOH) is rarely observed in human tumors and, consistent with prior reports[34], we see that variant allele frequency (VAF) does not differ between *NRAS* codon 61 mutations that are rare or enriched in human melanoma (Supplementary Fig. 4). *NRAS*

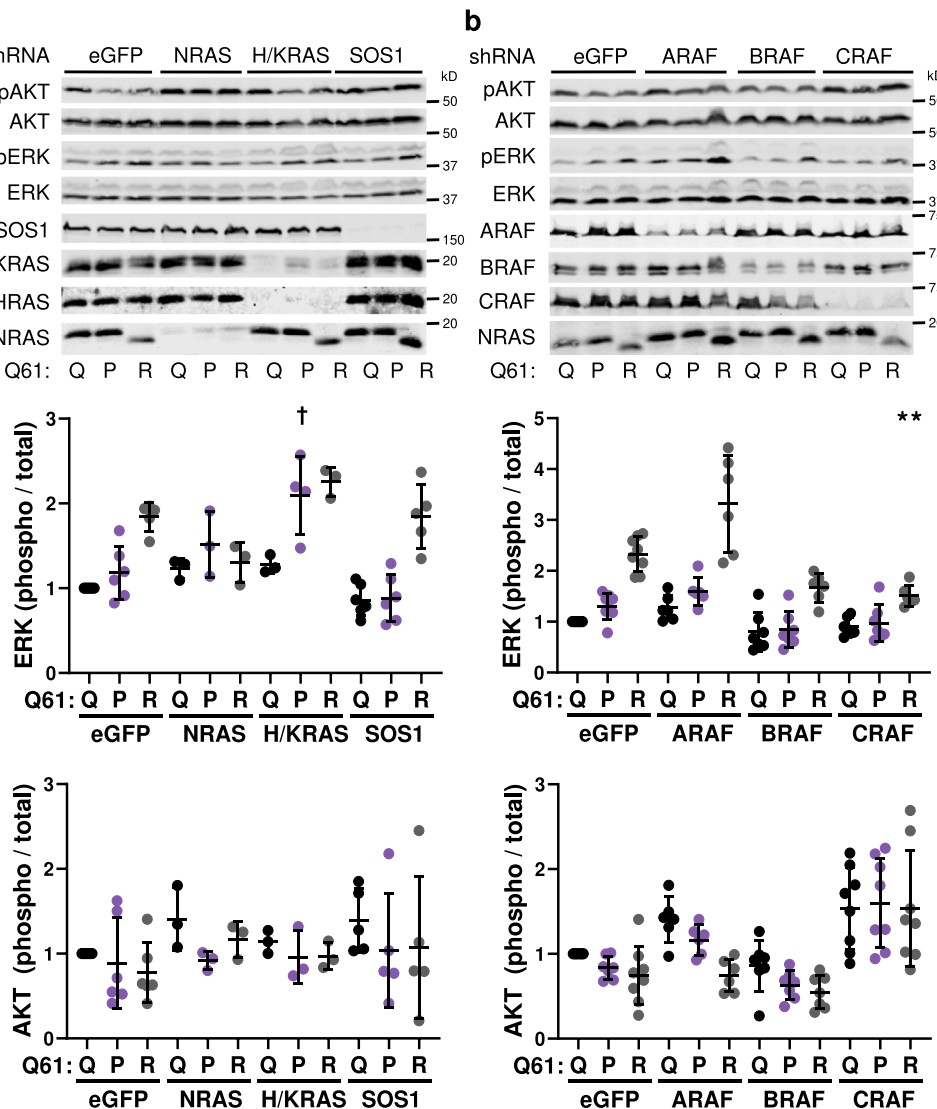

**Fig. 5 Oncogenic NRAS mutants mediate differential MAPK activation via a RAF-dependent mechanism.** Representative immunoblots of AKT and ERK activation in homozygous MEF cell lines treated with shRNAs targeting *Nras*, *Hras* and *Kras*, or *Sos1* (**a**) or *Araf*, *Braf* or *Craf* (**b**). Dot plot data are presented as mean values +/− SD where each dot represents one biological replicate. For **a** the following biologically independent replicates per genotype were examined over five independent experiments (eGFP arm: Q = 5, P = 5, R = 5; NRAS arm: Q = 3, P = 3, R = 3; H/KRAS arm: Q = 3, P = 4, R = 3; SOS1 arm: Q = 5, P = 5, R = 5). For **b** the following biologically independent replicates per genotype were examined over 8 independent experiments (eGFP arm: Q = 8, P = 8, R = 8; ARAF arm: Q = 6, P = 6, R = 6; BRAF arm: Q = 7, P = 7, R = 6; CRAF arm: Q = 7, P = 7, R = 6). Adjusted *p*-values were generated using a one-way ANOVA with a Tukey's multiple comparisons test. Statistics denoted in the figure indicate significant differences between shRNA-treated NRAS mutant MEFs and their respective eGFP control. ** *p* < 0.01, † *p* < 0.001. A complete list of adjusted *p*-values can be found in Supplementary Table 4a, b. Source data are provided as a Source Data file.

amplification is, however, more common in human melanomas with an NRAS codon 12 or 13 mutant, supporting our observations that endogenous NRAS codon 12 and 13 mutants are insufficient to drive melanomagenesis (Supplementary Figs. 2, 4). Future studies, in which a conditional *Nras* knockout mouse is crossed to the *TN*[61R] model, will be needed to fully address whether gene dosage is an important determinant of NRAS melanomagenic potential.

Wild-type RAS may also influence the evolutionary selection of RAS mutants in cancer. Specifically, Westcott et al. found that urethane-treated *Kras* homozygous and heterozygous mice develop lung tumors with distinct *Kras* mutations (Q61R and Q61L, respectively; ref. [23]). These data suggest that the presence of wild-type RAS may influence the evolutionary selection of RAS mutations in cancer. Here we saw that NRAS[61R] could not initiate

melanoma in the presence of wild-type NRAS (Fig. 2e). However, we did not investigate whether a single *Nras*[61R] allele has the potential to drive spontaneous melanoma formation or if wild-type NRAS can prevent tumor initiation by melanomagenic mutants other than NRAS[61R]. It remains possible that unrecognized polymorphisms linked to the *LSL-Kras*[G12D] allele promote the selection of KRAS 61L over 61R mutants in the Westcott studies. Finally, structural and functional differences between K- and NRAS may exert distinct evolutionary pressures in lung and skin tumorigenesis. Future in vivo analyses may also reveal a mutant-specific impact of wild-type NRAS on melanoma initiation.

Our data provide a mechanistic explanation for the selection of NRAS mutants in melanoma. We used computational modeling to show that melanomagenic NRAS mutants populate conformers

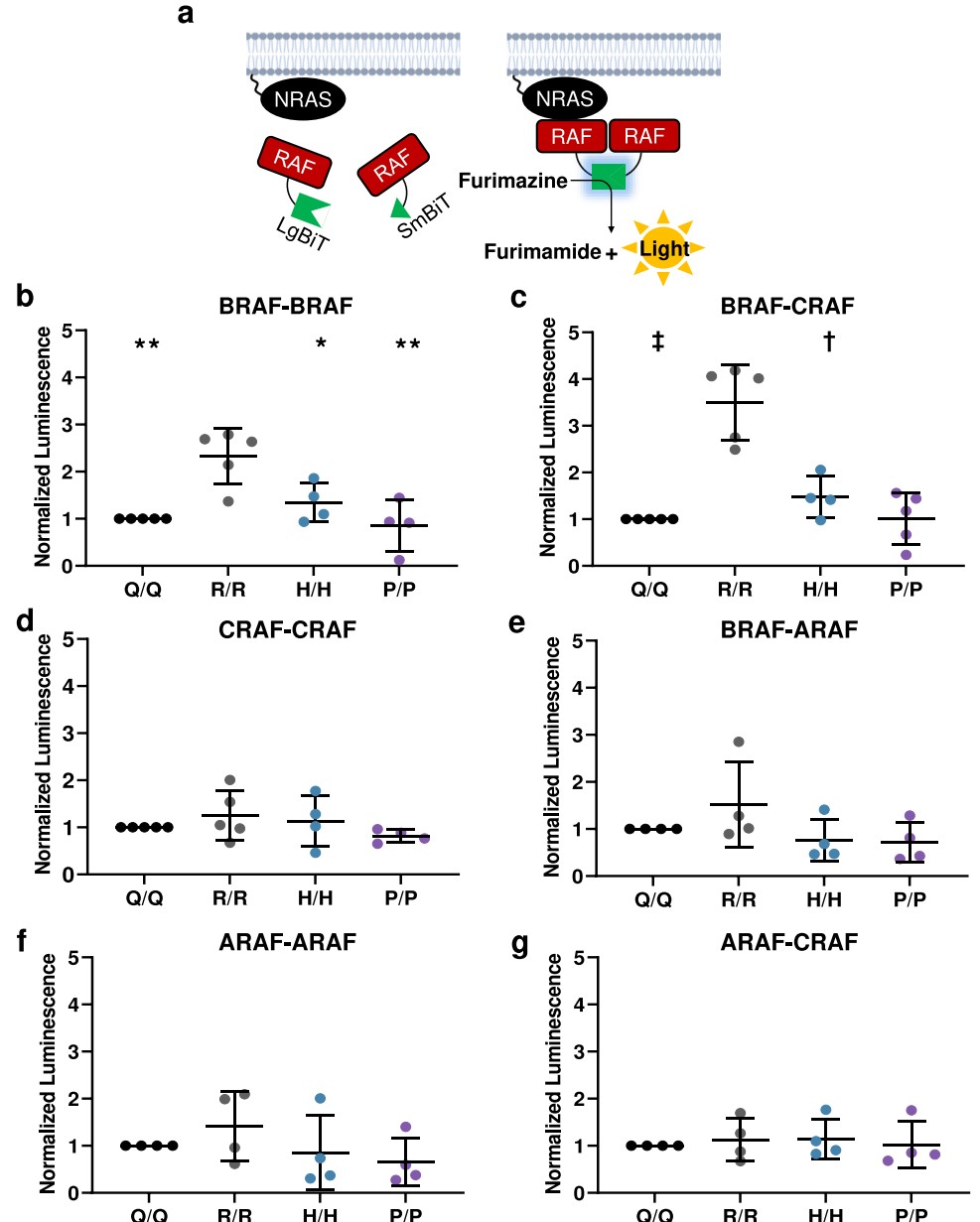

**Fig. 6 Melanomagenic NRAS mutants enhance RAF dimerization. a** Schematic representation of the RAF NanoBiT assay in which each RAF isoform is tagged with either LgBiT or SmBiT. Dot plots of normalized luminescence intensity in $TN^{61X/X}$ MEFs infected with adenovirus expressing BRAF-LgBiT and BRAF-SmBiT (**b**), BRAF-LgBiT and Craf-SmBiT (**c**), CRAF-LgBiT and CRAF-SmBiT (**d**), ARAF-LgBiT and BRAF-SmBiT (**e**), ARAF-LgBiT and ARAF-SmBiT (**f**), or ARAF-LgBiT and CRAF-SmBiT (**g**). Luminescence intensity was normalized to crystal violet staining for each well. Dot plot data are presented as mean values +/− SD where each dot represents one biological replicate. The following biologically independent replicates per genotype were examined over five independent experiments (BRAF-BRAF: Q/Q = 5, R/R = 5, H/H = 4; P/P = 4; BRAF-CRAF: Q/Q = 5, R/R = 5, H/H = 4; P/P = 5; CRAF-CRAF: Q/Q = 5, R/R = 5, H/H = 4; P/P = 4; BRAF-ARAF: Q/Q = 4, R/R = 4, H/H = 4; P/P = 4; ARAF-ARAF: Q/Q = 4, R/R = 4, H/H = 4; P/P = 4; ARAF-CRAF: Q/ Q = 4, R/R = 4, H/H = 4; P/P = 4). One-way ANOVA with a Tukey's post-test was used to compare data between each genotype. NRAS mutant samples statistically different from NRAS⁶¹ᴿ/ᴿ samples are indicated in the figure. Adjusted $p$-values for all comparisons can be found in Supplementary Table 4d. * $p < 0.05$, ** $p < 0.01$, † $p < 0.001$, ‡ $p < 0.0001$. Source data are provided as a Source Data file.

amenable to BRAF binding (Fig. 7a, b; Supplementary Fig. 12). Live-cell NanoBiT and BRET assays confirmed that melanomagenic NRAS mutants bind and activate BRAF better than non-melanomagenic NRAS mutants (Figs. 6, 7c, d; Supplementary Figs. 10, 11). Moreover, preference for the formation of BRAF dimers was observed in both MEFs and melanocytes expressing a melanomagenic NRAS mutant (Fig. 6; Supplementary Figs. 10, 11). BRAF-CRAF heterodimers increased more than any other RAF dimer in cells expressing melanomagenic NRAS mutants. Since the catalytic activity of BRAF-CRAF heterodimers exceeds

that of either homodimer[35], these data suggest that such a pairing would optimally enhance MAPK > ERK signaling. MAPK signaling plays a pivotal role in human melanoma evolution, with increased activity occurring early in tumor onset and strengthening throughout disease progression[36]. Thus, our findings provide a mechanism by which melanomagenic NRAS mutants achieve the levels of MAPK > ERK signaling required for tumorigenesis.

The idea that higher MAPK > ERK signaling favors melanomagenesis is supported by human and murine data. For example,

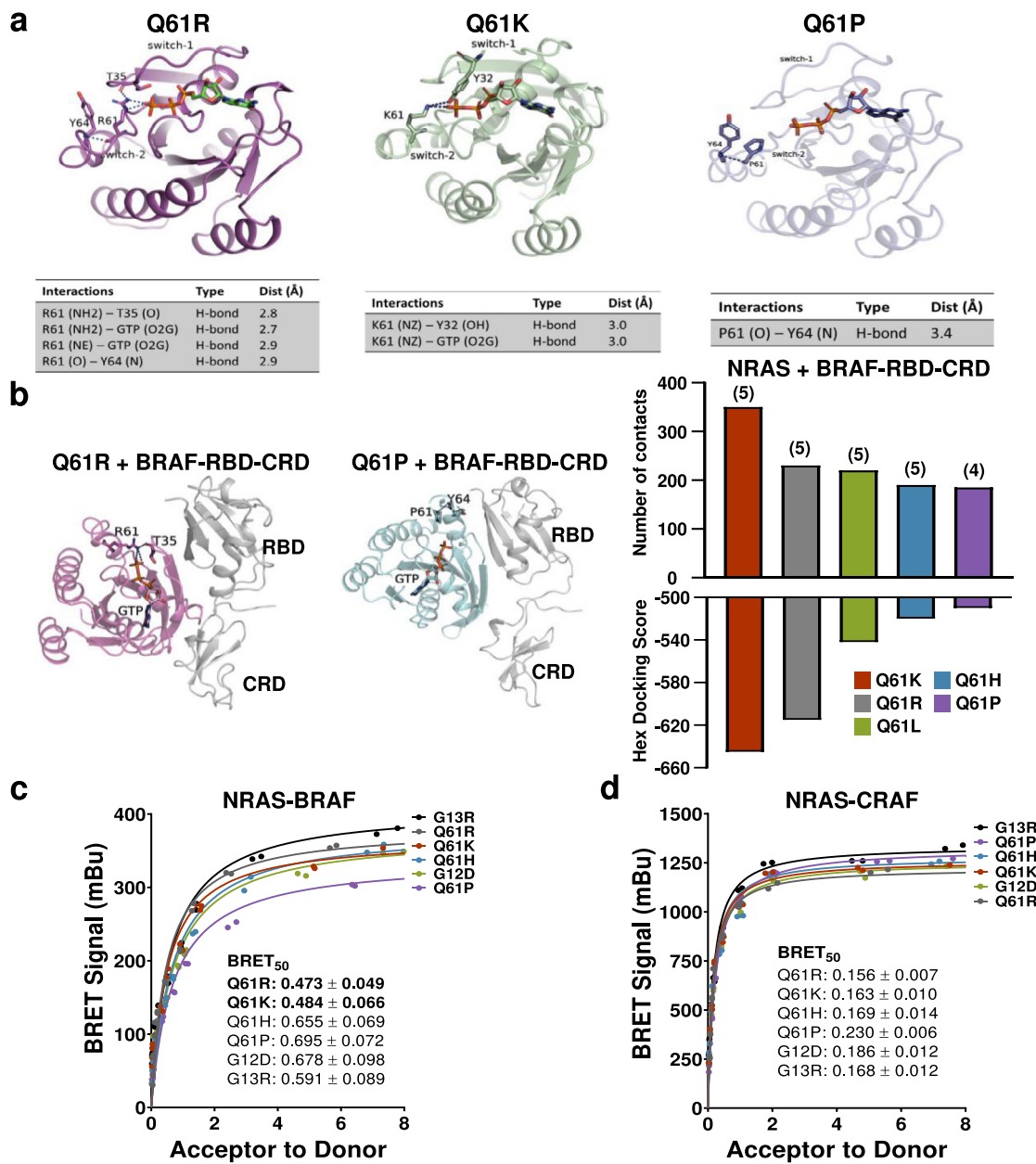

**Fig. 7 Conformational changes induced by NRAS mutants alter BRAF binding affinity. a** Representative conformations of NRAS[61R], NRAS[61K], and NRAS[61P] extracted from their highly populated replica-exchange molecular dynamics (REMD) structural ensembles. Interactions with the codon 61 sidechain are listed below each structure. **b** Binding orientation of NRAS[61R] and NRAS[61P] with the BRAF-RBDCRD as generated using Hex molecular docking simulations. The average conformation representing highly populated structural ensembles extracted from each NRAS codon 61 mutant trajectory was docked against the BRAF-RBDCRD. In the cartoon representation, the NRAS codon 61 mutant and bound nucleotide are shown in licorice, the BRAF-RBDCRD in gray and polar interactions for each mutant, and its surrounding residues are indicated by blue dashed lines. Comparisons of the interaction energy and the number of contacts between the BRAF-RBDCRD and each NRAS mutant suggest that highly melanomagenic NRAS mutants (NRAS[61R], NRAS[61K]) bind BRAF with higher affinity than NRAS[61H], NRAS[61L], and NRAS[61P]. The number of autoinhibitory contacts relieved by NRAS mutant binding is listed in parentheses. BRET protein–protein interaction data from Venus-tagged NRAS mutant and Rluc8-tagged BRAF (**c**) or CRAF (**d**) constructs co-transfected into 293T cells at increasing receptor to donor ratios. The data shown are representative of two replicates. Best fit BRET$_{50}$ values (binding affinity) and standard error, determined by non-linear regression, are shown for each mutant. Bolded values indicate statistically significant values as compared to both NRAS[61H] and NRAS[61P]. *p*-values determined by t-tests with 20 degrees of freedom representing the number of measures per curve. Source data are provided as a Source Data file.

non-melanomagenic mutants, such as NRAS[12D], are commonly detected in combination with *NRAS* amplification or activating mutations in other components of the MAPK pathway[37,38]. Similarly, when a non-melanomagenic mouse model expressing NRAS[12D] is crossed to a kinase-dead *BRAF* allele capable of inducing paradoxical RAF activation, melanomagenesis ensues[39].

Here we observed that NRAS expression was dramatically elevated in melanomas containing a relatively weak driver, NRAS[61H] (Fig. 4b). These observations further support the idea that weaker activators of the MAPK > ERK pathways likely require additional genomic alterations to initiate melanoma. Ultraviolet radiation may also facilitate melanomagenesis by stimulating the release of

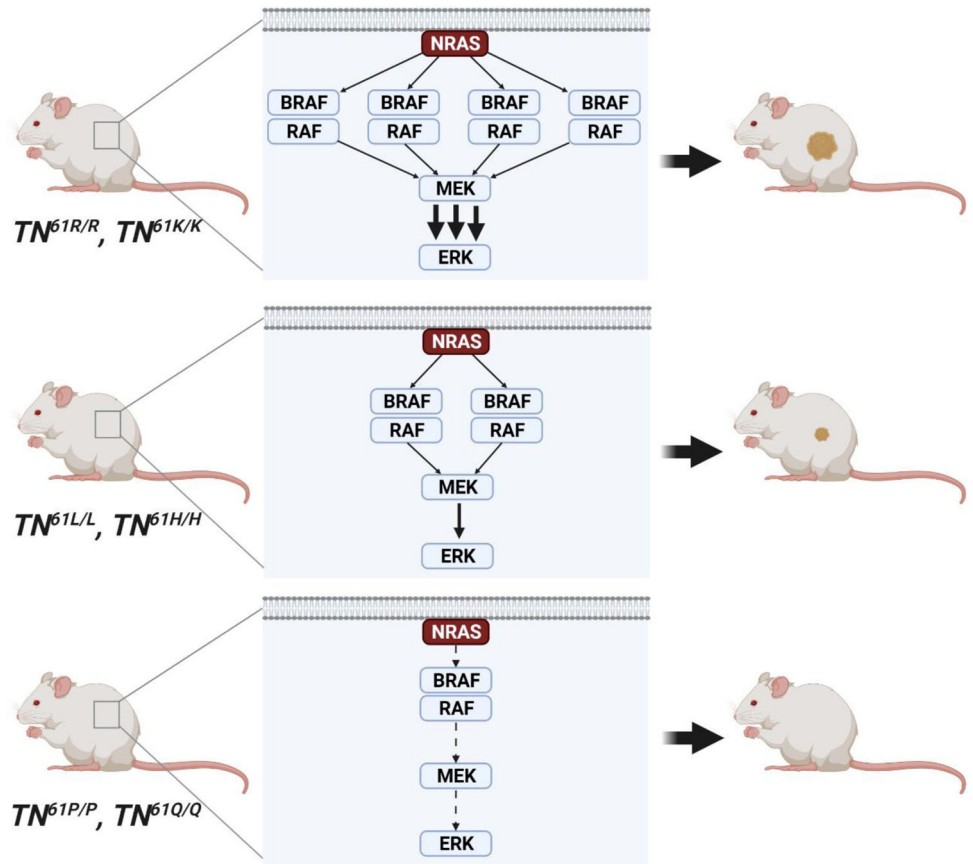

**Fig. 8 Differential RAF engagement explains variances in the ability of oncogenic NRAS mutants to initiate melanoma formation.** Image created with BioRender.com.

paracrine growth factors that augment MAPK > ERK signaling by simultaneously activating wild-type RAS[40]. Nevertheless, the fact that negative regulators of the MAPK pathway are elevated both in our mouse models and in human melanomas (Supplementary Fig. 4) makes it clear that MAPK signaling must be carefully balanced during disease onset. Perturbing this balance, in one direction or the other, could be key to melanoma prevention.

Our data support a mutant- and disease-specific approach to targeting RAS-driven cancers. Complete blockade of NRAS[61R] or NRAS[61K] may not be necessary if the functional properties of these alleles could be shifted toward a phenotype or conformation that resembles NRAS[12D] or NRAS[61P]. Our findings suggest that limiting NRAS-BRAF interactions could prevent the formation of NRAS-mutant melanoma. The versatile suite of inducible, endogenous *Nras* alleles we describe should enable the broader scientific community to identify and target mutant-specific requirements for RAS tumorigenesis in other tumor types.

## Methods

**Murine alleles and husbandry**. Animal work was performed in compliance with protocols approved by The Ohio State Institutional Care and Use Committee (Protocol #2012A00000134). Animals are housed in temperature (72.5 °F) and humidity (48.9%) controlled rooms with a 12 h light cycle (lights on from 6 am to 6 pm). The *LSL-Nras[61R]* allele and *TN* model were previously backcrossed >7 generations to C57BL/6J[6] (MMRRC #043604-UNC). Other *LSL-Nras[61X]* alleles were created via zygotic gene editing with CRISPR-Cas9 technology (gRNA and homology oligo sequences provided in Supplementary Table 4a). Codon 61 alleles were generated from C57BL/6J *TN[61R/R]* homozygous zygotes, whereas codon 12 and codon 13 alleles were generated from C57BL/6J *TN[61Q/Q]* homozygous zygotes. Targeting was verified in the resulting offspring by Sanger Sequencing (primers provided in Supplementary Table 5a). During this process, a silent G/A mutation was discovered in the 3rd nucleotide of codon 15 of the *LSL-Nras[12D]* and *LSL-*

*Nras[13R]* alleles. Each allele was backcrossed two generations to *TN[61R/R]* mice prior to beginning experiments.

**In vivo Cre induction and UV exposure**. NRAS expression was initiated by applying 20 mM 4-hydroxytamoxifen (4-OHT) to the backs of neonatal pups on postnatal days one and two[6]. On postnatal day three, animals were subjected to a single, 4.5 kJ/m² dose of ultraviolet B (UVB) using a fixed position 16 W, 312 nm UVB light source (Spectronics #EB-280C). [See ref. [18] for additional information]. Experimental cohorts included both male and female mice.

**Outcome monitoring and histopathology**. Mice from each cohort were randomly numbered and blindly monitored three times a week for tumor formation. Upon detection, melanomas were measured three times per week and tumor size (width × length (mm)) was recorded using calipers. Mice were euthanized upon reaching any of the pre-determined exclusion criteria which included: a single tumor of ≥1.6 cm in any dimension, >1 tumor with any one tumor being ≥1.3 cm in diameter, tumor ulcerations of >2 mm in size, or body condition of <2/5[41]. Careful tracking of each experimental mouse was performed to ensure that the maximum tumor size was not exceeded. Mice were euthanized by CO₂ inhalation followed by cervical dislocation in accordance with the guidelines of the American Veterinary Medical Association. A portion of each primary tumor was fixed in 10% neutral buffered formalin and the rest was flash-frozen for protein extraction. Formalin-fixed samples were paraffin-embedded, sectioned (4 μm), and stained with hematoxylin and eosin (H&E). Stained tumor sections were evaluated using an Olympus BX45 microscope with an attached DP25 digital camera (B&B Microscopes Limited, Pittsburgh, PA) by a veterinary pathologist certified by the American College of Veterinary Pathologists (K.M.D.L.).

**Immunohistochemistry**. Tumors were fixed overnight in 10% neutral buffered formalin and embedded in paraffin. Tumor sections (5 μm) were deparaffinized in xylene and rehydrated in ethanol. Antigen retrieval was performed in a steamer with Dako Antigen Retrieval Solution (#S1699) for 30 min. The tumor sections were then blocked with 3% hydrogen peroxide, avidin, and biotin and were incubated overnight in primary antibody directed against Cre Recombinase (1:125, Cell signaling #15036S). Slides were treated with IHC Select biotinylated secondary antibody (Goat anti-rabbit IgG, Millipore #21537) and VectaStain R.T.U Elite ABC Reagent (Vector Labs #PK-7100). Next, DAB chromogen was added to each section

for 30 s. The tissue was blocked a second time in 2.5% normal horse serum (Vector Laboratories #S-2012-50) and was incubated with a second primary antibody directed against Ki67 (1:1000, Abcam, #264429), CD45 (1:200, Cell Signaling #70257S), or Cleaved-Caspase-3 (1:250, Cell signaling #9579S). Slides were treated with horse anti-rabbit IgG secondary antibody (ImPRESS AP Horse Anti-Rabbit IgG Polymer Kit, Vector Laboratories MP-5401) and stained with Fast-Red Substrate kit (Abcam, #ab64254) for 12 min. Lastly, the slides were counterstained with hematoxylin diluted 1:10 in PBS for 23 s. Three representative images of each slide were taken on an Olympus BX53M brightfield microscope with an SC30 camera attachment. Percentages of DAB and FastRed positive cells were quantified using the ImageJ (version 1.53 m) Colocalization Object Counter plugin (version 1.0.0)[42].

**Isolation and culture of primary mouse embryonic fibroblasts and immortalized melanocytes.** MEFs were generated from E13.5 embryos using manual homogenization and trypsinization. Dissociated cells were cultured in fibroblast growth medium (Dulbecco's modified eagle medium (DMEM), supplemented with 10% fetal bovine serum (FBS), 1% penicillin-streptomycin, and 1% glutamine). MEF lines were passaged when confluency reached 70–80% in a 10 cm tissue culture dish.

To generate primary melanocyte lines, neonatal mice were euthanized and their skin was subjected to mechanical and enzymatic homogenization with digestion buffer that includes 10% FBS, 1% penicillin/streptomycin solution, 1% L-glutamine, 10 mg/mL collagenase type I, 0.25% porcine trypsin and 0.02 mg/mL deoxyribonuclease I in RPMI 1640[43]. Homogenized cells were then plated on a collagen-coated 6 cm plate in melanocyte base medium (Ham's F12 containing 10% FBS, 7% Horse Serum (Thermo Fisher 26050088)) with growth supplements (0.5 mM di-butyryl cyclic AMP (dbcAMP; Sigma D0627), 20 nM phorbol 12-myristate 13-acetate (TPA; Sigma P8139), 200pM cholera toxin (Sigma, C8052), 1% penicillin-streptomycin and 1% glutamine). Once a pure population of melanocytes was established, the cells were immortalized by CRISPR/Cas9-mediated targeting of *trp53*. Here, adenovirus containing constructs expressing Cas9 and a *trp53* guide RNA (seed sequence: 5′- GTGTAATAGCTCCTGCATGG) were added to primary melanocytes cultured in serum-free Ham's F12 media. Eight hours post-infection, primary melanocytes were washed with PBS and placed in melanocyte base medium for recovery. Immortalized melanocyte lines were passaged upon reaching 60–70% confluency in a 6 cm tissue culture dish.

**In vitro induction of NRAS expression.** MEFs and primary melanocytes were seeded at equal densities in 6 or 10 cm tissue culture plates. The following day, these cultures were washed with PBS and placed in fibroblast or melanocyte base medium. Adenovirus expressing Cre recombinase conjugated to eGFP (Ad5-CMV-Cre-eGFP; Baylor College of Medicine Vector Development Laboratory, Houston, TX) was added to the cultures for 16 h (MEFs) or 8 h (melanocytes) at an MOI of 4000:1 (viral particles: cells). After infection, cells were allowed to recover for at least 72 h in fibroblast or melanocyte growth medium prior to analysis. Allelic recombination was confirmed through genomic PCR in which one of three possible PCR products were possible: wild-type (487 bp), LSL-Nras (371 bp), or CRE recombined *LSL-Nras* (562 bp). The following PCR primers were used for genotyping recombination in these alleles: Primer 1—5′-AGACGCGGAGACTTGGCGAGC-3′ (0.15 μmol/L); Primer 2—5′-GCTGGATCGTCAAGGCGCTTTTCC-3′ (0.15 μmol/L); Q61R GENO2—5′-GCAAGAGGCCCGGCAGTACCTA-3′ (0.15 μmol/L). The samples were run under the following cycling conditions 95 °C 15 min, 35 × [94 °C 30 s, 62 °C 30 s, 72 °C 45 s], 72 °C 5 min[6].

**Immunoblotting.** Frozen tumors (10–15 mg) were homogenized using a liquid nitrogen-cooled mortar and pestle. Homogenized tumor tissue and pelleted cell lines were lysed in RIPA (25 mM Tris pH 7.4, 150 mM NaCl, 1% IGEPAL, 0.1% SLS) supplemented with protease inhibitor cocktail (Sigma P8340), calyculin A (CST 9002S) and Halt phosphatase inhibitor cocktail (Thermo Fisher 78420). Equal protein concentrations, were determined by Bradford Assay (Bio-Rad #5000006), were run on an SDS-PAGE gel and transferred to PVDF (Sigma IPFL00010). PVDF membranes were blocked in 5% milk-PBS and then probed with one of the following primary antibodies: ERK1/2 (1:1000, CST 4696S), phospho-ERK1/2 (1:1000, CST 9101S), AKT (1:1000, CST 2920), phospho-AKT (1:1000, CST 9271), NRAS (1:250, Abcam ab77392), HRAS (1:1000, Abcam ab32417), KRAS (1:1000, Sigma WH0003845M1), SOS1 (1:1000, CST 5890), ARAF (1:1000, CST 4432P), BRAF (1:500, Santa Cruz sc-5284), or CRAF (1:500, CST 12552). Secondary antibodies were diluted in 5% BSA 1x PBST as follows: anti-goat (1:15000. LI-COR 926-32214), anti-mouse (1:15000, LI-COR 926-68070, LI-COR 926-32210), or anti-rabbit (1:15000, LI-COR 926-68071, LI-COR 926-32211). Membranes were imaged on a LI-COR Odyssey CLx system and quantified using Image Studio software (LI-COR Biosciences).

**RNA-Sequencing.** NRAS expression was induced in passage 3 MEFs using Ad5-CMV-Cre-eGFP as described for our in vitro studies. The cells were then cultured for 6 days prior to RNA isolation using the ZR-Duet DNA/RNA MiniPrep Plus Kit (Zymo D7003). The ZR-Duet DNA/RNA MiniPrep Plus Kit (Zymo D7003) was also used to isolate RNA from *TN61X/X* tumor tissue. In brief, a mortar and pestle were pre-chilled with liquid nitrogen before frozen tissue was added to the mortar

along with liquid nitrogen. After the liquid nitrogen evaporated, the frozen tissue was ground into a fine powder, mixed with Zymo DNA/RNA Lysis Buffer, then processed as described in the ZR-Duet DNA/RNA MiniPrep Plus Kit. RNA quality and concentration were confirmed on an Agilent TapeStation and Life Technologies Qubit. RNA was prepared for sequencing through ribosomal depletion using Illumina Ribo-Zero chemistry followed by library preparation using Illumina TruSeq Total RNA Stranded Library Prep Kit. RNA was sequenced on an Illumina HiSeq4000 or NovaSeq6000 with 150 base-pair, paired-end reads. Raw data files are deposited in the NCBI Gene Expression Omnibus (GEO) under accession #GSE162124 (MEFs) or # GSE197841 (tumor samples).

RNA reads were aligned to build 38 of the mouse genome (mm10) using STAR[44], duplicates marked using PICARD (version 2.17.11) (http://broadinstitute.github.io/picard/) and a gene count matrix generated by featureCounts (version 1.22.2)[45]. Differential gene expression analysis was performed using DESeq2 (p-adjusted < 0.05)[46]. GSEA used the DOSE algorithm within the GSEA function of the clusterProfiler package[47,48] to probe gene sets from the molecular signatures database Hallmark collection[49]. Gene Ontology (GO) analysis was performed using the "enrichGO" algorithm in the clusterProfiler package in R[47,48].

**Whole exome sequencing.** Tumor DNA was isolated from flash-frozen tissue using the same mortar and pestle technique described for RNA-seq and a *Quick*-DNA MiniPrep Plus Kit (Zymo D4068). Frozen tissue was ground into a fine powder and mixed with 400uL of Zymo Solid Tissue Buffer. Samples were then incubated at 55 °C for 2 h before continuing with the Solid Tissue protocol outlined in the *Quick*-DNA MiniPrep Plus Kit. DNA concentration was confirmed on a Life Technologies Qubit. Exome enrichment was performed by Novogene using the Agilent SureSelectXT Mouse Exon Kit and sequenced on a NovaSeq 6000 with 150 base-pair, paired-end reads. Raw data files are deposited in the Sequence Read Archive (SRA) under bioproject #PRJNA812398.

Sequencing reads were aligned to build 38 of the mouse genome (mm10) using burrows-wheeler aligner (version 0.7.15)[50], duplicates were removed using PICARD (version 2.17.11) (http://broadinstitute.github.io/picard/), and the reads were realigned around indels using GATK version 3.6[51]. Variants were called using Mutect2[52], VarScan2 (version 2.4)[53], and Strelka2[54]. Variants that were not detected by all three algorithms or present in the Ensembl mouse variation database were filtered out of each dataset[55]. Filtered datasets were annotated with Variant Effect Predictor[56]. Total mutational burden and the prevalence of Signature 7 were determined using *MutationalPatterns*[57].

**Flow cytometric analysis of EdU labeling.** Passage three *TN61X/X* MEFs were infected with Ad5-CMV-Cre-eGFP to induce NRAS expression as described for our in vitro studies. MEFs were then cultured for five days. MEFs were then incubated in DMEM containing 1% penicillin-streptomycin and 1% glutamine for five hours prior to adding 0.01 mM 5-ethynyl-2-deoxyuridine (EdU) to the media. MEFs were labeled with EdU for an additional five hours and then harvested and fixed with 4% paraformaldehyde. Fixed cells were permeabilized with saponin in 1% BSA 1× PBS. Click-iT chemistry was used to label the incorporated EdU with Chromeo 642. Here, the cells were incubated for 30 min in Click-iT reaction cocktail containing 2 mM CuSO4, 50 mM ascorbic acid, and 50 nM Chromeo 642 azide dye (Active Motif 15288) diluted in 1× PBS. 10,000 cells per sample were analyzed on a BD LSR Fortessa flow cytometer and the percentage of EdU positive cells was determined using FlowJo software. Specifically, the initial population of MEFs was selected by gating based on FSC-A by SSC-A (Supplementary Fig. 7, top). Next, cell doublets were removed by gating for single cells in a FSC-H by FSC-A plot (Supplementary Fig. 7, middle). Finally, a histogram of counts by APC-A intensity was used to determine the percent of EdU positive cells in each population of MEFs (Supplementary Fig. 7, bottom).

**EdU labeling of melanocytes in vivo.** Neonatal pups were induced to express NRAS and treated with UVB irradiation as described above. EdU (0.041 mg/kg) was administered to mice on postnatal day 10 via intraperitoneal injection. Two hours later, mice were euthanized and the dorsal skin was collected. Samples were fixed in 10% neutral buffered formalin for 24 h, embedded in paraffin, and cut into 5 μm sections. Slides of each section were deparaffinized and rehydrated and then Click-iT chemistry was used to label the incorporated EdU with Chromeo 642 as described above. Antigen retrieval was performed using Dako Antigen Retrieval Solution (Agilent S169984-2) followed by blocking with Dako Protein Block (Agilent X090930-2). Cutaneous melanocytes were labeled with anti-gp100 primary antibody (1:100; Abcam ab137078) and Alexa Fluor 555 secondary antibody (4 μg/mL, Thermo Fisher A21428). Nuclei were counterstained with DAPI (1:10,000). Five images were taken for each biological replicate on a Perkin Elmer Vectra automated quantitative pathology imaging system and each image was counted by five blinded reviewers.

**shRNA knock-downs in MEFs.** Mission shRNA vectors purchased from Sigma were transiently transfected along with pCMV-VSVG and ps-PAX2 into HEK 293T (ATCC #CRL-3216) cells using polyethylenimine (PEI) at a ratio of 3 μL of 10 μg/μL PEI per 1 μg of plasmid (shRNA information provided in Supplementary Table 5b). HEK 293T cells are validated and authenticated by short tandem repeat

(STR) analysis and undergo mycoplasma testing on a yearly basis. Viral supernatant was collected 48- and 72 h post-transfection and filtered through a 0.45 μm syringe filter. Viral supernatant was added to NRAS-null MEFs along with 10 μg/mL polybrene. Fresh media was placed on the cells the following day and 1.5 μg/mL puromycin selection began 48 h post-infection.

**qPCR analysis in primary melanocytes**. *NRAS* expression was induced in primary melanocytes using Ad5-CMV-Cre-eGFP as described for our in vitro studies. The cells were then cultured for 6 days prior to RNA isolation using the ZR-Duet DNA/RNA MiniPrep Plus Kit (Zymo D7003). RNA concentration was confirmed on an Agilent TapeStation and Life Technologies Qubit. cDNA was prepared from 200 ng RNA using the iScript Select cDNA Synthesis Kit (Biorad 1708897). qPCR reactions were prepared using 2 μL of a 1:4 dilution of cDNA mixed with 350 nM target-specific primers (Supplementary Table 5c) and SensiFAST-mix SYBR Hi-ROX (Bio-line 92020). qPCR reactions were run under normal cycling conditions with 40 cycles (95 °C for 10 s, 61 °C for 15 s, 72 °C for 10 s). A melt curve was used to confirm a single amplified product for each primer pair. Gene expression levels were assessed by calculating $2^{-\Delta\Delta Ct}$ values normalized to wild-type NRAS[61Q] control.

**Adenoviral amplification**. RAF NanoBiT and *trp53* gRNA sequences were cloned into pAdTrack[58]. *Pme1*-linearized pAdTrack plasmid was then electroporated into BJ5183-AD-1 cells. The recombinant AdEasy vector[58] isolated from the transformed cells was digested with *Pac1* and transfected into HEK 293AD cells (Agilent #240085) using polyethylenimine (PEI) at a ratio of 30 μg PEI to 1 μg of plasmid. HEK 293AD cells are validated and authenticated by short tandem repeat (STR) analysis and undergo mycoplasma testing on a yearly basis. Following serial propagation of the virus through HEK 293AD cells, adenovirus was purified using a CsCl gradient and dialyzed in dialysis buffer (10 mM Tris (pH 8), 2 mM MgCl, 4% sucrose). The purified virus was mixed with glycerol and stored at −80 °C.

**NanoBiT assays**. Passage four $TN^{61X/X}$ MEFs or immortalized melanocytes were treated with Ad5-CMV-Cre to induce NRAS expression and then equally seeded into a 96 well plate. The following day, the cells were placed in fibroblast or melanocyte medium with low serum and infected with adenovirus expressing the indicated RAF NanoBiT constructs. The following day, the cells were washed in PBS and placed in the appropriate growth medium for recovery. Forty-eight hours post-infection, the cells were washed with PBS and incubated in serum-free DMEM or RPMI containing 1% penicillin-streptomycin and 1% glutamine for four hours prior to analysis. Luminescence intensity was assessed using the Nano-Glo Live Cell Assay (Promega N2012). The cells were then fixed in 10% neutral buffered formalin with crystal violet (0.01% w/v) for 30 min. Crystal violet-stained plates were imaged on a LI-COR CLx and quantified with Image Studio software. Luminescence intensity was normalized to the crystal violet staining intensity for each well.

**Replica-exchange molecular dynamics (REMD) simulations**. We employed REMD simulations to sample protein conformations of various NRAS mutants[27]. As structural information is missing for NRAS-Q61K/H/L/P mutants in the RCSB database, we performed molecular modeling to generate starting structures for MD simulations. We mutated GppNHp-bound wildtype NRAS (PDB: 5UHV[59],) to generate NRAS-Q61H/L/P mutant structures using the Pymol *mutagenesis* tool. We chose the side chain rotamers of H61, L61, and P61 with minimal clash score and modified GppNHp to GTP through appropriate modifications in the triphosphate tail. Since Arg and Lys residues have similar biophysical properties, we used GTP-bound NRAS-Q61R (PDB: 6ZIZ[60],) as a template to model the NRAS-Q61K structure. All missing residues and atoms were relocated using Modeler-9v18 tool[61] prior to MD simulations.

The CHARMM36 forcefield[62,63] was used to generate the topology of NRAS and bound GTP nucleotide. NRAS Q61K/H/L/P mutant structures and the NRAS Q61R X-structure were separately solvated (after removing bound CRAF RBD-CRD) in a periodic water box with an appropriate number of Na+ and Cl− counterions to maintain 150 mM salt concentration. Each system was energy minimized by employing 10,000 steps of steepest-descents followed by another 10,000 steps of conjugate-gradients algorithms. Subsequently, position restrained equilibration simulations were performed on each NRAS Q61 mutant system in isothermal–isobaric ensemble (constant temperature and pressure) for 10 nanoseconds. V-rescale thermostat[64] and Parrinello-Rahman barostat[65] were used to maintain temperature and pressure at designated values. Electrostatic interactions were evaluated using the particle mesh Ewald method with a cutoff distance of 1.2 nm. van der Waals interactions were terminated at a cutoff value of 1.2 nm and LINCS algorithm was used to constrain all bonds with H-atoms. REMD simulations on 32 replicas were performed within the temperature range 290–350 K using GROMACS-2020.3[66]. Temperatures of individual replicas were generated using the temperature generator web server (http://folding.bmc.uu.se/remd/)[67]. Exchange trials among 32 replicas were performed for every 2 ps with an exchange rate of 0.25. Altogether, the REMD simulations were executed on each NRAS-Q61 mutant for 9.6 μs. Clustering analysis was performed on each REMD system to identify representative structural ensembles. All structural figures were rendered using Pymol visualization software (The PyMOL Molecular Graphics System, Version 2.0 Schrödinger, LLC.).

**Molecular docking**. The recent X-structure of the CRAF:KRAS-Q61R complex (PDB 6XGU[68],) shows numerous contacts between RAS and the RAS binding domain-cysteine rich domains (RBD-CRD) of CRAF. Although a cryo-EM reconstruction of BRAF in both autoinhibited and RAS-bound states was recently solved[69], the RBD-CRD structure was not observable in the RAS-bound complex. Hence, we used active CRAF RBD-CRD crystal structure to model BRAF interactions with NRAS Q61 mutants by homology modeling. The optimized structure of the BRAF RBD-CRD was then used to estimate interactions with NRAS-Q61 variants by protein–protein docking approaches. Representative, highly populated structural ensembles of NRAS-Q61K/R/H/L/P mutants were docked to the RBD-CRD region of BRAF using the Hex docking program[70]. Hex uses real orthogonal spherical polar basis functions to represent surface shape and charge distributions of receptors and ligands. Hex employs FFT calculations to estimate probable docked complex conformations and docking scores of the protein–protein complex as a function of the six degrees of translational and rotational freedom in a rigid body docking search. Based on Hex estimations, we identified high-affinity binding complexes, as evident from the Hex docking score, and evaluated inter-protein interactions.

**Bioluminescence resonance energy transfer (BRET) assays**. Analysis of RAS-RAF interactions using the BRET assay was performed as previously described[71]. In brief, Venus-tagged NRAS and Rluc8-tagged RAF were co-transfected into 293T cells at an increasing ratio of 1:0.05 − 1:8 (Rluc:Venus) wherein 62.5 ng Rluc8-tagged RAF was consistently transfected. Live cells were collected 48 h post-transfection and BRET signal was assessed 2 min following the addition of coelenterazine-h to the cells. Non-linear regression was used to plot the best fit hyperbolic curve from two saturation curves. These curves were then used to determine $BRET_{50}$ values. Best fit $BRET_{50}$ values and standard error are shown for each mutant. Statistical significance was determined by t-tests with 20 degrees of freedom representing the number of measures per curve.

**Statistics and reproducibility**. Statistical analyses for Kaplan–Meier curves and dot plots were performed using GraphPad Prism version 8.4.3. Survival differences in Kaplan–Meier curves were assessed using log-rank (Mantel–Cox) tests. One-way ANOVA was used to compare conditions in each dot plot and correct for multiple comparisons as stated in each figure legend. Dot plots depict the mean ± s.d. of data acquired from ≥3 biological replicates with each dot representing a single replicate. $p < 0.05$ was considered significant.

**Reporting summary**. Further information on research design is available in the Nature Research Reporting Summary linked to this article.

## Data availability

The raw RNA sequencing data are available on NCBI Gene Expression Omnibus under the following accession numbers: GSE162124 (MEFs) or GSE197841 (tumor samples). The raw whole exome sequencing data are available on NCBI Sequencing Read Archive under bioproject number: PRJNA812398. RNA sequencing and whole exome sequencing data were aligned to build 38 of the mouse genome (mm10). Data from Supplementary Fig. 4 were obtained from cBioPortal MSKCC Melanoma (cBioPortal for Cancer Genomics: NRAS in Melanoma (MSKCC, Clin Cancer Res 2021)), TCGA PanCancer Atlas (cBioPortal for Cancer Genomics: NRAS in Pan-cancer analysis of whole genomes (ICGC/TCGA, Nature 2020)), and TCGA Cancer Cell Line datasets (cBioPortal for Cancer Genomics: NRAS in Cancer Cell Line Encyclopedia (Novartis/Broad, Nature 2012) and 2 other studies)[37,38]. Data for Supplementary Fig. 6 were obtained from the UCSC Xena platform (https://doi.org/10.1038/s41587-020-0546-8) (UCSC Xena (xenabrowser.net)). The remaining data are available within the Article, Supplementary Information, or Source Data file.

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

## Acknowledgements

The authors thank the members of the Genetically Engineered Mouse Modeling Core (GEMMC) at The Ohio State University and the Genomics Services Laboratory (GSL) at Nationwide Children's Hospital for their technical services. In addition, we would like to thank Emma Crawford, Makanko Komara, and Suohui Zhang for technical support, Dr. Krista M. D. LaPerle for pathology support, Dr. Dafna Bar-Sagi (NYU) for reagents, and Dr. Martin McMahon (University of Utah) for critical reading of the manuscript. This work was supported by the Damon Runyon Foundation (38-16; C.E.B.), NIH F31 CA236418 (B.M.M.), Pelotonia (B.M.M.), NIH T32 GM068412 (B.M.M.), NIH P30 CA016058 (OSUCCC), NCI ZIA BC 010329 (D.K.M), NIH R35 GM134962 and P01 CA203657 (S.L.C.), and American Cancer Society PF-20-140-01-CDD (L.M.C.).

## Author contributions

B.M.M. and C.E.B. conceived the study and wrote the manuscript. M.C. and V.C. performed CRISPR-Cas9 targeting to develop the NRAS-mutant TN mouse models. B.M.M., C.E.B., T.J.W., A.M.H., and M.F. generated the experimental mouse colonies, tracked tumor formation, and contributed to the analysis of in vivo data. B.M.M. isolated primary cells and performed in vitro signaling assays with help from R.E.L., A.D. and M.S.B. V.P. and R.E.L. performed and quantified IHC analysis of murine tumor samples. B.M.M. and C.J.B. processed and analyzed the RNA sequencing data. E.M.T. and D.K.M. performed and analyzed the BRET assays. V.R.C., L.M.C., and S.L.C. executed and contributed to the analysis of all molecular simulation data. All authors assisted in editing the manuscript.

## Competing interests

C.E.B. and B.M.M. are inventors of TN61 cell lines licensed to Millipore. All other authors declare no competing interests.
