## [Peer Review File · Nature Communications]

Enhanced BRAF engagement by NRAS mutants capable of promoting melanoma initiationReviewers' Comments:

Reviewer #1:

Remarks to the Author:

In this paper, Murphy and colleagues describe tumor formation and signaling in a panel of NRAS mutants, with the goal of understanding the increased prevalence of NRAS Q61 mutants in melanoma. In some respects this is an extension of earlier work from Burd and colleagues that showed that G12D mutations in NRAS fail to form melanomas in strains of mice in which NRAS Q61 mutations are efficient at doing so. This study remains one of the few clear examples of allele specific tumor formation.

Through a careful and comprehensive analysis of a cohort of mouse mutants, Murphy and colleagues make a number of interesting observations. They address the dominance of wild type NRAS over the NRAS Q61R mutant, for example, as well as effects at combining alleles that have not been reported previously. This dominant effect has not been reported previously for NRAS, though it has for HRAS and KRAS, as cited. Wescott and colleagues showed that KRASQ61R alleles do not co-occur with WT KRAS, whereas KRASQ61L do. This paper is relevant and should be cited here. In addition, it would be useful to know if different alleles co-exist with wild type alleles in human melanoma samples.

The molecular basis of the differences reported here are not clear. The distinction between codon 12 and codon 61 mutants is discussed briefly (lines 222, 223), but this comparison is not strictly accurate. Both types of mutation are resistant to GAP-mediated GTP hydrolysis, but codon 61 mutations are also inhibited for intrinsic GTP hydrolysis. Whether this is relevant to these distinctions is unknown, however. Also, there seems to be a nice correlation between biological activity and activation of the MAPK pathway amongst Q61 alleles. The inference might be that G12 mutants are weaker in this capacity, but this was not measured. The fact that Q61 mutations do not activate PIK kinase in these cells interesting. Is this also true of G12D? These comparisons may be beyond the scope of this paper but should be discussed.

Analysis of RAF dependence and dimerization is interesting but not very surprising. Overall, this paper represents a step forward in characterizing these important mutant forms of RAS. Unfortunately, it does not contain any new insights as to why the different alleles should differ in their transforming power. Their differential activation of RAF dimers is a step in this direction, but it is not clear why these alleles would have this effect. Are they all loaded with GTP to the same extent? Do they bind RAF isoforms differently, or do have distinct structural features, as suggested by Buhrman and Mattos many years ago?? Obviously these are far beyond the scope of this paper, but the title of the paper does suggest some new insights into functional differences that are not addressed here. The paper does establish that codon frequencies reflect properties of the protein rather than mutational exposure, though this point is not made very strongly. For example, Hahn and colleagues have shown that RAS codon frequencies do tend reflect mutational burden with a few exceptions.

In conclusion, this a nice paper that describes interesting biological features of an important panel of mutants, and makes some interesting advances in the field. It is weakened by lack of molecular analysis to explain the basis of these effects.

Reviewer #2:

Remarks to the Author:

The manuscript by Murphy, Burd and colleagues describes the generation and analysis of a new, high-quality panel of GEM models with different configurations of mutant NRAS. The strong positive of this work is the creation of these models, and the excellent assessment of their in vivo potential to provoke melanomas, using a variety of configurations and combinations, as well as +/- UV, although the later was not pursued in any significant way. It is clear these models will be a valuable resource to the field in the years ahead. Much of the data presented were consistent with what is know or observed in patients with melanoma.

The negative in this study is that mechanisms were not followed up from early leads, and the

preponderance of these mechanistic studies were based on the analysis of MEFs from the new GEM lines generated, the authors assuming that these results would apply to melanocytic cells. This assumption is not always valid, and the appropriate cell type should be used to validate key conclusions. It is not clear with all the effort that went into making these GEMs with various NRAS alleles why cell lines were not generated for the most important melanoma classes. These would represent ideal mechanistic reagents for analysis and would also be invaluable to the field.

Following up on this, why did the authors not perform full transcriptomic analyses on the primary melanomas with different genotypes? These could provide relevant data to try to confirm what is observed in MEFs, as well as give you highly relevant *in vivo* data. These data would greatly strengthen the paper.

For the MEF transcriptomics, once RNA differences were noted between melanomagenic vs. non-melanomagenic NRAS configurations [61R/R vs. 61P/P, and 61H/H and 61P/P (Fig. 3A)], it seems a reasonable strategy to compare the “winners” from these two separate analyses. Both the common and distinct changes could very likely help prioritize key factors/pathways to pursue more vigorously.

The melanomas derived from various genotypes show no difference in tumor growth *in vivo*, yet MEFs and cutaneous melanocytes show differences in EdU incorporation (proliferation). What is the proliferation in the melanoma cells within the various tumors? Are there differences that are masked by apoptosis or other mechanisms to maintain steady tumor growth? Again, this gets to *in vivo* relevance.

OVERVIEW

We appreciate the opportunity to address prior Editorial and Reviewer concerns. These insightful suggestions strengthened the rigor, scope, and impact of our findings. Our revised manuscript establishes a fundamental model to explain how melanomagenic NRAS mutants drive enhanced MAPK>ERK signaling to facilitate melanoma progression. New structural, biochemical, genomic, and cell culture data support our discovery that melanomagenic NRAS mutants bind BRAF with greater affinity than non-melanogenic NRAS mutants. Our work explains why certain NRAS mutants are enriched in human melanoma, and the common overlap between tumor types driven by NRAS Q61R/K and BRAF. These findings also highlight how mutation-specific changes in RAS structure and dynamics can result in distinct effector interactions, downstream signaling, and biology. A point-by-point response to each reviewer's critique follows below.

REVIEWER 1

R1 Q1: In some respects this is an extension of earlier work from Burd and colleagues that showed that G12D mutations in NRAS fail to form melanomas in strains of mice in which NRAS Q61 mutations are efficient at doing so.

We agree that it is not surprising that NRAS codon 12 and 13 mutants were poor drivers of melanoma given our prior publication, the distinct biochemical properties of NRAS codon 12, 13, and 61 mutants, and the paucity of codon 12 and 13 alterations in human melanoma. However, this manuscript establishes that different melanoma phenotypes result from distinct substitutions at the same amino acid. Furthermore, new molecular dynamics simulations, molecular modeling, biochemical, genomic, and cell culture data included in our revised submission uncover the mechanism by which specific NRAS mutants differentially promote enhanced MAPK>ERK signaling and melanoma progression. (See R1 Q5.)

These data confirm that skin-specific mutagenic processes are not the reason why certain NRAS codon 61 mutants are more frequently observed in human melanoma. Additionally, our findings highlight the importance of RAS switch domain dynamics in dictating the downstream consequences of mutant RAS expression.

R1 Q2: [The dominant effect of wild-type NRAS over NRAS Q61R] has not been reported previously for NRAS, though it has for HRAS and KRAS, as cited. Wescott and colleagues showed that KRASQ61R alleles do not co-occur with WT KRAS, whereas KRASQ61L does. This paper is relevant and should be cited here. It would [also] be useful to know if different [NRAS] alleles co-exist with wild-type alleles in human melanoma samples.

We appreciate this suggestion and have included the following text, figures, and references in our revised manuscript.

Discussion lines 190-211:

“Our analysis of heterozygous *TN* mice revealed an interesting, additive effect of mutant, but not wild-type, NRAS on melanomagenesis. Prior studies show that a single, wild-type RAS allele can limit the tumorigenic potential of RAS mutants of the same isoform. This observation is supported by data from several human tumor types in which loss or downregulation of the cognate wild-type allele is frequent. In line with these findings, our results reveal that NRAS^{61R} cannot initiate melanoma formation in the presence of a wild-type allele (Manuscript Figure 2E). However, NRAS^{61R} retains the ability to initiate melanoma when expressed in combination with a non-melanomagenic, GTPase defective NRAS^{61P} allele (Manuscript Figure 2D). NRAS loss of heterozygosity (LOH) is rarely observed in human tumors and, consistent with prior reports, we see that variant allele frequency (VAF) does not differ between NRAS codon 61 mutations that are rare or enriched in human melanoma (Manuscript Figure S4, **Rebuttal Figure R1**). NRAS amplification is, however, more common in human melanomas with an NRAS codon 12 or 13 mutant, supporting our observations that endogenous NRAS codon 12 and 13 mutants are insufficient to drive melanomagenesis (Manuscript Figure S2, S4, **Rebuttal Figure R1**). Future studies, in which a conditional *Nras* knockout mouse is crossed to the *TN*^{61R} model, will be needed to fully address whether gene dosage is an important determinant of NRAS melanomagenic potential.

Wild-type RAS may also influence the evolutionary selection of RAS mutants in cancer. Specifically, Westcott et al. found that urethane-treated *Kras* homozygous and heterozygous mice develop lung tumors with distinct *Kras* mutations (Q61R and Q61L, respectively). These data suggest that the presence of wild-type RAS may influence the evolutionary selection of RAS mutations in cancer. Here we saw that NRAS^{61R} could not initiate melanoma in the presence of wild-type NRAS (Manuscript Figure 2E). However, we did not investigate whether a single *Nras*^{61R} allele has the potential to drive

spontaneous melanoma formation or if wild-type NRAS can prevent tumor initiation by melanomagenic mutants other than NRAS^{61R}. It remains possible that unrecognized polymorphisms linked to the *LSL-Kras*^{G12D} allele promote the selection of KRAS 61L over 61R mutants in the Westcott studies. Finally, structural and functional differences between K- and NRAS may exert distinct evolutionary pressures in lung and skin tumorigenesis. Future in vivo analyses could also reveal a mutant-specific impact of wild-type NRAS on melanoma initiation.”

Figure R1 (Manuscript Figure S4): Loss-of-heterozygosity is rare in NRAS-driven malignancies. A-C, NRAS copy number alterations (left) and variant allele frequencies (right) in NRAS-mutant melanomas from the MSKCC Melanoma dataset (A), NRAS-mutant malignancies from the TCGA PanCancer Atlas (B), and NRAS-mutant cell lines from the TCGA Cancer Cell Line dataset (C). In the dot blots at the right, each tumor is represented by a single dot. Error bars show the mean and standard deviation of each genotype.

R1 Q3: The molecular basis of the differences reported here are not clear. The distinction between codon 12 and codon 61 mutants is discussed briefly (lines 222, 223), but this comparison is not strictly accurate. Both types of mutation are resistant to GAP-mediated GTP hydrolysis, but codon 61 mutations are also inhibited for intrinsic GTP hydrolysis. Whether this is relevant to these distinctions is unknown, however.

Our revised manuscript provides molecular dynamics simulation data and biochemical assays to explain the mechanism by which melanomagenic NRAS mutants drive enhanced MAPK>ERK signaling (See response to **R1 Q5**). We appreciate the Reviewer's point about the differences between codon 12 and 61 mutants and have modified the statement referenced above to read: "However, these results might be predicted because codon 61 mutants have a more profound effect on RAS intrinsic GTPase activity ." (Manuscript lines 183-184).

R1 Q4: ...There seems to be a nice correlation between biological activity and activation of the MAPK pathway amongst Q61 alleles. The inference might be that G12 mutants are weaker in this capacity, but this was not measured. The fact that Q61 mutations do not activate PI3K kinase in these cells is interesting. Is this also true of G12D?

In response to this query, we compared MAPK>ERK signaling and RAF dimerization in cells expressing endogenous levels of each NRAS codon 12, 13, or 61 mutant. ERK phosphorylation and RAF dimerization were elevated only in the presence of the melanomagenic NRAS codon 61 mutants (**Rebuttal Figure R2**). Furthermore, increased MAPK>ERK signaling in MEFs expressing NRAS G13R as compared to MEFs expressing NRAS G12D and G13D, coincides with the weak melanomagenic potential of NRAS G13R (**Rebuttal Figure R2A-B**, Manuscript Figure S9A-B).

We also measured PI3K activation downstream of NRAS codon 12, 13, and 61 mutants using AKT S473 phosphorylation as a readout (**Rebuttal Figure R2A**, Manuscript Figure S9A&C). Each NRAS mutant drove a similar level of PI3K activity, suggesting that these mutants exhibit comparable levels of AKT activation when expressed at endogenous levels.

Figure R2: Poor induction of RAF dimerization, MAPK>ERK and PI3K>AKT signaling by endogenous NRAS codon 12 and 13 mutants. **A, (Manuscript Figure S9A)** *Left:* Immunoblot of protein lysates isolated from MEFs expressing the indicated endogenous NRAS mutants. *Right:* Dot plot of normalized ERK p42/44 and AKT S473 phosphorylation across replicates. Each dot represents one biological replicate. ANOVA with a Dunnett T3 multiple comparisons test was used to compare data from each NRAS mutant to MEFs expressing NRAS^{61R/R}. Adjusted p-values for all comparisons can be found in Manuscript Table S5B. ** p< 0.01, † p< 0.0001. **B-G, (Manuscript Figure S11)** Dot plot of normalized luminescence intensity in *TN^{XX}* MEFs infected with adenovirus expressing BRAF-LgBiT and BRAF-SmBiT (B), BRAF-LgBiT and CraF-SmBiT (C), CRAF-LgBiT and CRAF-SmBiT (D), ARAF-LgBiT and BRAF-SmBiT (E), ARAF-LgBiT and ARAF-SmBiT (F) or ARAF-LgBiT and CRAF-SmBiT (G). Luminescence intensity was normalized to crystal violet staining for each well. ANOVA with a Dunnett T3 multiple comparisons test was used to compare normalized luminescence intensity in MEFs expressing each NRAS mutant to MEFs expressing NRAS^{61R}. Adjusted p-values for all comparisons can be found in Manuscript Table S6E. * p< 0.05, ** p< 0.01, † p< 0.001, ‡ p< 0.0001.

R1 Q5: (This paper) does not contain any new insights as to why the different alleles should differ in their transforming power. Their differential activation of RAF dimers is a step in this direction, but it is not clear why these alleles would have this effect. Are (all mutants) loaded with GTP to the same extent? Do they bind RAF isoforms differently, or do have distinct structural features, as suggested by Buhrman and Mattos many years ago??

We hypothesized that structural variations in the NRAS Q61 mutants alter BRAF affinity based on published data mentioned by the Reviewer^{5,6}. In collaboration with Dr. Sharon Campbell's lab, we focused on the switch I and II regions of RAS, known to mediate RAS-RAF interactions. We discovered that the melanomagenic NRAS Q61R and Q61K mutants sample conformations with improved affinity for BRAF (**Rebuttal Figure R3A-B**, Manuscript Figure 7, S12A-F). We further collaborated with Dr. Deborah Morrison's group to test this model by measuring the relative affinity of each NRAS mutant for BRAF and CRAF in live cells (**Rebuttal Figure R3C-D**, Manuscript Figure 7C-D). Using BRET complementation assays, we discovered that NRAS Q61R and Q61K have a greater affinity for BRAF than non-melanomagenic NRAS mutants. However, all NRAS mutants showed a similar affinity for CRAF. These molecular modeling and biochemical data provide a new,

fundamental model to explain why melanomagenic NRAS mutants exhibit enhanced BRAF engagement, activation, and downstream signaling.

Figure R3: (Manuscript Figure 7) Conformational changes induced by NRAS mutants alter BRAF binding affinity. A, Representative conformations of NRAS^{61R}, NRAS^{61K} and NRAS^{61P} extracted from their respective highly populated REMD structural ensembles. Interactions with the codon 61 sidechain are listed below each structure. **B,** Binding orientation of NRAS^{61R} and NRAS^{61P} with the BRAF-RBDCRD as generated using Hex molecular docking simulations. The average conformation of each NRAS codon 61 mutant was extracted from REMD simulation trajectories and subsequently docked against the BRAF-RBDCRD. In the cartoon representation, the NRAS codon 61 mutant and bound nucleotide are shown in licorice, the BRAF-RBDCRD in gray, and polar interactions of each mutant and its surrounding residues are indicated by blue dashed lines. Comparisons of the interaction energy and number of contacts between the BRAF-RBDCRD and each NRAS mutant suggest that highly melanomagenic NRAS mutants (NRAS^{61R}, NRAS^{61K}) bind BRAF with higher affinity than NRAS^{61H}, NRAS^{61L} and NRAS^{61P}. The number of autoinhibitory contacts relieved by NRAS mutant binding is listed in parenthesis. **C-D,** BRET protein-protein interaction data from Venus-tagged NRAS mutant and Rluc8-tagged BRAF (C) or CRAF (D) constructs co-transfected into 293T cells at increasing receptor to donor ratios. Best fit BRET₅₀ values (binding affinity) and standard error, determined by non-linear regression, are shown for each mutant. Bolded values indicate statistically significant values as compared to both NRAS^{61H} and NRAS^{61P}. p-values determined by t-tests with 20 degrees of freedom representing the number of measures per curve.

R1 Q7: Obviously these are far beyond the scope of this paper, but the title of the paper does suggest some new insights into functional differences that are not addressed here. The paper does establish that codon frequencies reflect properties of the protein rather than mutational exposure, though this point is not made very strongly. For example, Hahn and colleagues have shown that RAS codon frequencies do tend to reflect mutational burden with a few exceptions.

We appreciate the Reviewer's suggestion to highlight the key findings of our manuscript in both the title and discussion. The title of this manuscript is now "Enhanced BRAF engagement by NRAS mutants capable of promoting melanoma initiation".

We wished to further explore the Reviewer's comment about the relationship between mutational burden and RAS codon frequencies. Therefore, we performed whole-exome sequencing of genomic DNA isolated from three NRAS Q61R, Q61L, and Q61H tumors. Tumors containing weaker melanoma-driving mutants had a higher mutational load (**Rebuttal Figure R4A**), suggesting that additional hits were required for tumorigenesis. However, UV did not appear to be responsible for disparities in mutational load. In fact, the UV mutational signature, COSMIC Signature 7, was equally prevalent in NRAS Q61R, Q61L, and Q61H tumors (**Rebuttal Figure R4B**). These findings suggest that disparities in mutational load may reflect differences in the tumorigenic potential of RAS mutants. We mention these data in our revised Discussion (Manuscript lines 227-229).

Figure R4: A, Dot plot of total mutational burden in *TN* melanomas driven by the denoted NRAS mutants. Each dot represents an individual tumor. **B,** Stacked bar plot illustrating the percent contribution of the UV-associated mutational signatures (COSMIC Signature 7a-d) to the overall mutational profile in melanomas driven by the denoted NRAS mutants. * p < 0.05.

R2 Q1: The negative in this study is that mechanisms were not followed up from early leads, and the preponderance of these mechanistic studies were based on the analysis of MEFs from the new GEM lines generated, the authors assuming that these results would apply to melanocytic cells. This assumption is not always valid, and the appropriate cell type should be used to validate key conclusions.

Primary melanocytes are extremely difficult to establish and grow in culture for several reasons: 1) Primary melanocytes require a compendium of growth factors and rapidly senesce in culture; and 2) Primary melanocytes from C57BL/6 mice cannot be cryopreserved due to high levels of melanin. To circumvent these challenges, we immortalized melanocytes from each of our mouse models via CRISPR/Cas9-mediated *Trp53* knockout. These lines were then used to validate key mechanistic findings from our original submission, including the enhanced MAPK>ERK signaling, gene regulation patterns, and RAF dimerization associated with melanomagenic NRAS mutants in MEFs (**Rebuttal Figure R5**, Manuscript Figure S6, S10).

In addition to these experiments, we performed structural, biochemical, and cell-based assays to refine our understanding of how melanomagenic and non-melanomagenic NRAS mutants differ mechanistically (See **Rebuttal Figure R3**). These data indicate that NRAS Q61R and Q61K adopt structures distinct from other non-melanomagenic mutants. These structural differences alter intramolecular switch I and II interactions, leading to the enhanced affinity of melanomagenic NRAS mutants for the BRAF RBD-CRD domain. Thus, differences in intramolecular switch I and II interactions lead to distinct and biologically significant functional outcomes, which could someday be modulated to curb RAS oncogenicity.

Figure R5: Enhanced MAPK>ERK signaling and RAF activation in melanocytes expressing endogenous melanomagenic NRAS mutants. **A**, (Manuscript Figure S10A) Immunoblot of protein lysates isolated from *TN* primary melanocytes expressing the indicated endogenous NRAS mutants. Below each blot is the normalized intensity of pERK^{p42/44} / ERK and pAKT^{S473} / AKT in relation to the NRAS Q61Q sample. **B-F**, (Manuscript Figure 6B) Bar plots of relative *ETV4* (B), *CCND1* (C), *DUSP6* (D), *SPRY2* (E), or *SPRY4* (F) mRNA expression in primary *TN* melanocytes expressing the indicated NRAS mutants. **G-I**, (Manuscript Figure S10B-D) Dot plot of normalized luminescence intensity in TN61X/X primary melanocytes infected with adenovirus expressing BRAF-LgBiT and BRAF-SmBiT (G), BRAF-LgBiT and CraF-SmBiT (H), or CRAF-LgBiT and CRAF-SmBiT (I). Luminescence intensity was normalized to crystal violet staining for each well. ANOVA with a Dunnet T3 multiple comparisons test was used to compare luminescence intensity from each codon 61 mutant to wildtype (TN^{61Q/Q}) melanocytes. Adjusted p-values for all comparisons can be found in Table S6C. * p< 0.05, ** p< 0.01, ‡ p< 0.0001.

R2 Q2: It is not clear with all the effort that went into making these GEMs with various *NRAS* alleles why cell lines were not generated for the most important melanoma classes. These would represent ideal mechanistic reagents for analysis and would also be invaluable to the field.

We agree that the generation of melanoma cell lines of differing genotypes would be of great value to the field. In fact, we have generated 13 melanoma lines from our *LSL-Nras* models. We have conducted extensive genomic, antigenic, and phenotypic characterizations of these lines and plan to publish a separate manuscript similar to the one describing the BRAF-mutant YUMM lines⁷. These lines have already been licensed to Millipore and will be offered publicly in 2022. Regarding the use of these cell lines in mechanistic analyses, please see our responses to **R2 Q3 & Q4**.

R2 Q3: (W)hy did the authors not perform full transcriptomic analyses on the primary melanomas with different genotypes?

We assumed previously that tumor RNA-Seq would not identify differences between melanomagenic and non-melanomagenic *Nras* alleles because most of the non-melanomagenic NRAS mutants did not form tumors. Furthermore, melanomas that did form in our mice had similar growth kinetics regardless of the underlying NRAS mutant. However, we performed RNA sequencing of six melanomas from each of the following GEMM genotypes: 61K, 61R, 61L, and 61H. Fewer than 32 genes were differentially expressed between tumors containing strong (Q61K/R) and moderate (Q61L) drivers of melanoma, whereas 761 genes were differentially expressed in NRAS Q61R and Q61H tumors (**Rebuttal Figure R6A-C**, Manuscript Figure S8). Gene Ontology analyses identified GTPase and GEF activity among the top biological processes differing between NRAS Q61R- and Q61H-driven melanomas (**Rebuttal Figure R6E**, Manuscript Figure S8). These data suggest that increases in NRAS expression could facilitate tumorigenesis in melanocytes carrying a weak driver like NRAS Q61H. Supporting this idea, our data show that NRAS is frequently overexpressed in Q61H-driven murine melanomas (**Manuscript Figure 4B**). Furthermore, increased MAPK signaling is a hallmark of human melanoma progression⁸ and Richard Marais' group has shown a non-melanomagenic *Nras* allele (G12D) can initiate murine melanoma when paired with a kinase-dead BRAF (D594A) mutant that induces paradoxical MAPK>ERK signaling⁹. These new and published data support our model that increased BRAF engagement is required for melanoma initiation and are highlighted in the revised manuscript (lines 223-235).

These transcriptomic data also revealed increased immune-associated genes in melanomas driven by NRAS Q61R versus melanoma driven by NRAS Q61H (**Rebuttal Figure R6D**). However, only 50% of Q61H tumors surveyed had a lower percentage of CD45⁺ immune infiltrates than Q61R tumors in IHC staining (**Rebuttal Figure R8B**). Thus, it is unclear if these gene signatures have a consistent biological impact on tumor development.

Figure R6: (Manuscript Figure S8) Transcriptomic alterations between strong and weak drivers of melanoma center around RAS GTPase binding and immune regulation. A-C, Volcano plot of differentially expressed genes in *TN* melanomas expressing NRAS^{61K/K} versus NRAS^{61R/R} (A), NRAS^{61L/L} versus NRAS^{61R/R} (B), or NRAS^{61H/H} versus NRAS^{61R/R} (C). **D,** Bar plot showing the enrichment of Hallmark gene sets (p-adjusted < 0.05) in melanomas expressing NRAS^{61H/H} versus NRAS^{61R/R}. **E,** Dot plot of the differentially regulated (p-adjusted < 0.05) molecular functions from Gene Ontology (GO) analysis in melanomas expressing NRAS^{61H/H} versus NRAS^{61R/R}.

R2 Q4: For the MEF transcriptomics, once RNA differences were noted between melanomagenic vs. non-melanomagenic NRAS configurations [61R/R vs. 61P/P, and 61H/H and 61P/P (Figure 3A)], it seems a reasonable strategy to compare the “winners” from these two separate analyses. Both the common and distinct changes could very likely help prioritize key factors/pathways to pursue more vigorously.

We appreciate this suggestion to bolster our analysis of our RNA sequencing data by identifying those pathways similarly upregulated by melanomagenic NRAS mutants (Q61R and Q61H) as compared to a non-melanomagenic NRAS mutant (Q61P). Thus, out of the 966 genes that were similarly altered between NRAS 61R MEFs and NRAS 61H MEFs versus NRAS 61P MEFs (**Rebuttal Figure R7A**), we focused on the 510 transcriptional targets upregulated in both NRAS 61R- and NRAS 61H-expressing MEFs. Gene ontology (GO) analysis of these transcriptional targets identified guanine nucleotide binding and GTPase activity as a molecular function enriched by melanomagenic NRAS mutants, supporting our prior findings (**Rebuttal Figure 7B**, Manuscript Figure 3A).

Figure R7: Differential regulation of the RAS-Myc axis by melanomagenic and non-melanomagenic NRAS mutants. **A**, Venn diagram showing the overlap of genes differentially expressed between NRAS 61P/P versus 61R/R or 61H/H MEFs. **B**, (Manuscript Figure 3A) Dot plots representing the molecular functions subset of Gene Ontology (GO) analysis of genes that are upregulated (left) or downregulated (right) in NRAS^{61R/R} and NRAS^{61H/H} MEFs compared to NRAS^{61P/P} MEFs

R2 Q5: What is the proliferation in the melanoma cells within the various tumors? Are there differences that are masked by apoptosis or other mechanisms to maintain steady tumor growth?

We performed immunohistochemical staining to address this question in melanomas of the following homozygous NRAS genotypes: 61K, 61R, 61L, and 61H. Proliferation rates were more elevated in *TN*^{61R/R}, *TN*^{61K/K} and *TN*^{61L/L} than *TN*^{61H/H} melanomas (Rebuttal Figure R8A-B). Immunohistochemical staining for CD45⁺ and cleaved Caspase showed considerable variability in immune-infiltration and apoptosis among tumors of different NRAS genotypes (Rebuttal Figure R8C). Thus, while we observe that NRAS Q61R increases cellular proliferation in vitro (Manuscript Figure 3D) and in vivo (Rebuttal Figure R8), it remains unclear why melanomas grow at similar rates regardless of the underlying *Nras* mutation present. In vivo measurements of furred mice are somewhat imprecise and may fail to capture subtle differences in growth rates.

Figure R8 (Manuscript Figure S1N-P, Table S1B): Quantification of immunohistochemistry (IHC) staining of *TN* melanoma tissue with Ki67 (A), CD45 (B), or cleaved Caspase 3 (C) antibodies. ANOVA with a Tukey's multiple comparison test was used to compare staining intensities between *TN* tissue sections. Adjusted p-values for all comparisons can be found in Table S1B. * $p < 0.05$ compared to *TN*^{61R/R}.

REVIEWER RESPONSE REFERENCES

- 1 Westcott, P. M. *et al.* The mutational landscapes of genetic and chemical models of Kras-driven lung cancer. *Nature* **517**, 489-492, doi:10.1038/nature13898 (2015).
- 2 Helias-Rodzewicz, Z. *et al.* Variation of mutant allele frequency in NRAS Q61 mutated melanomas. *BMC Dermatol* **17**, 9, doi:10.1186/s12895-017-0061-x (2017).
- 3 Adari, H., Lowy, D. R., Willumsen, B. M., Der, C. J. & McCormick, F. Guanosine triphosphatase activating protein (GAP) interacts with the p21 ras effector binding domain. *Science* **240**, 518-521, doi:10.1126/science.2833817 (1988).
- 4 Frech, M. *et al.* Role of glutamine-61 in the hydrolysis of GTP by p21H-ras: an experimental and theoretical study. *Biochemistry* **33**, 3237-3244, doi:10.1021/bi00177a014 (1994).
- 5 Buhrman, G., Kumar, V. S., Cirit, M., Haugh, J. M. & Mattos, C. Allosteric modulation of Ras-GTP is linked to signal transduction through RAF kinase. *J Biol Chem* **286**, 3323-3331, doi:10.1074/jbc.M110.193854 (2011).
- 6 Buhrman, G., Wink, G. & Mattos, C. Transformation efficiency of RasQ61 mutants linked to structural features of the switch regions in the presence of Raf. *Structure* **15**, 1618-1629, doi:10.1016/j.str.2007.10.011 (2007).
- 7 Meeth, K., Wang, J. X., Micevic, G., Damsky, W. & Bosenberg, M. W. The YUMM lines: a series of congenic mouse melanoma cell lines with defined genetic alterations. *Pigment Cell Melanoma Res* **29**, 590-597, doi:10.1111/pcmr.12498 (2016).
- 8 Shain, A. H. *et al.* Genomic and Transcriptomic Analysis Reveals Incremental Disruption of Key Signaling Pathways during Melanoma Evolution. *Cancer Cell* **34**, 45-55 e44, doi:10.1016/j.ccell.2018.06.005 (2018).
- 9 Pedersen, M., Viros, A., Cook, M. & Marais, R. (G12D) NRAS and kinase-dead BRAF cooperate to drive naevogenesis and melanomagenesis. *Pigment Cell Melanoma Res* **27**, 1162-1166, doi:10.1111/pcmr.12293 (2014).

Reviewers' Comments:

Reviewer #1:

Remarks to the Author:

The earlier version of this paper was of high quality in establishing different biological activities associated with Q61 alleles of NRAS, but lacked a mechanistic basis for this important observation. The authors now provide an interesting and convincing explanation that greatly increases the impact of this study. They have also addressed my other concerns in a very thorough way and I now recommend the paper for publication with enthusiasm.

Reviewer #2:

Remarks to the Author:

I wish to thank the authors for conducting the experiments that I had suggested. I found the results to be most interesting. And these and other modifications have greatly strengthened this manuscript.